# Amphipathic Cell-Penetrating Peptide-Aided Delivery of Cas9 RNP for In Vitro Gene Editing and Correction

**DOI:** 10.3390/pharmaceutics15102500

**Published:** 2023-10-20

**Authors:** Mert Öktem, Enrico Mastrobattista, Olivier G. de Jong

**Affiliations:** Department of Pharmaceutics, Utrecht Institute for Pharmaceutical Sciences (UIPS), Utrecht University, Universiteitsweg 99, 3584 CG Utrecht, The Netherlands; m.oktem@uu.nl (M.Ö.); e.mastrobattista@uu.nl (E.M.)

**Keywords:** CRISPR-Cas9, cell-penetrating peptide (CPP), LAH5, RNP, HDR, delivery

## Abstract

The therapeutic potential of the CRISPR-Cas9 gene editing system in treating numerous genetic disorders is immense. To fully realize this potential, it is crucial to achieve safe and efficient delivery of CRISPR-Cas9 components into the nuclei of target cells. In this study, we investigated the applicability of the amphipathic cell-penetrating peptide LAH5, previously employed for DNA delivery, in the intracellular delivery of spCas9:sgRNA ribonucleoprotein (RNP) and the RNP/single-stranded homology-directed repair (HDR) template. Our findings reveal that the LAH5 peptide effectively formed nanocomplexes with both RNP and RNP/HDR cargo, and these nanocomplexes demonstrated successful cellular uptake and cargo delivery. The loading of all RNP/HDR components into LAH5 nanocomplexes was confirmed using an electrophoretic mobility shift assay. Functional screening of various ratios of peptide/RNP nanocomplexes was performed on fluorescent reporter cell lines to assess gene editing and HDR-mediated gene correction. Moreover, targeted gene editing of the *CCR5* gene was successfully demonstrated across diverse cell lines. This LAH5-based delivery strategy represents a significant advancement toward the development of therapeutic delivery systems for CRISPR-Cas-based genetic engineering in in vitro and ex vivo applications.

## 1. Introduction

The CRISPR (Clustered Regularly Interspaced Short Palindromic Repeats)-Cas9 system has emerged as a flexible gene editing tool for both cell engineering and therapeutic applications [1,2,3,4,5]. For Cas9-mediated genome editing to be effective, different components need to be delivered inside the nucleus of target cells. These include the Cas9 protein, a guide RNA that targets the Cas9 nuclease to a specific sequence in the genomic DNA and, in case of gene correction or insertion, a DNA template to enable precise gene editing via homologous recombination. The sheer size of these components, combined with the need for the successful delivery of each component in optimal stoichiometric ratios, is the prime reason why effective delivery has proven to be difficult. Viral vectors can be constructed to carry genetic information to encode for the Cas9 protein as well as the single guide RNA as a means of intracellular delivery of CRISPR-Cas. However, because expression from viral promoters is often strong and long-lasting, this can give rise to unintended off-target effects and potential immunogenic responses [6,7,8]. More transient systems are therefore desired. 

Direct delivery of the Cas9:sgRNA ribonucleoprotein complex (RNP) offers several advantages. First, it is fast, as it does not depend on transcription, translation and subsequent assembly into an active RNP inside the cell, which can often be suboptimal due to different degradation profiles of the Cas9 protein and the sgRNA. Second, its transient nature reduces the chance of off-target cutting events and potential immune responses [9]. Third, it avoids potential risks of insertional mutagenesis, which is often associated with viral vectors [10].

To this end, different strategies have recently been explored for the delivery of Cas9 RNP into cells, including electroporation [11] and induced osmotic transduction (iTOP) [12], as well as the use of chemical transfection reagents [13] and non-viral vectors [9,14,15,16,17,18]. However, developing synthetic vectors for the direct delivery of CRISPR-Cas RNP with high efficiency has been challenging. Such synthetic vectors should avidly complex or encapsulate Cas9 RNP in order to protect it from premature proteolytic degradation, and it should trigger cellular uptake, endosomal escape and vector disassembly to enable the free RNP to be transported into the nucleus through the nuclear pore complexes. 

Cell-penetrating peptides (CPPs) have been explored for the delivery of Cas9 RNPs in various cell types [19,20,21,22,23]. CPPs are short (5–30 amino acids), polycationic or amphipathic peptides that facilitate the cellular uptake of different types of cargoes, such as small molecules, proteins and plasmid DNA, as well as nanoparticles [6,24]. The mechanism of action of CPPs is not fully understood but is generally believed to follow two uptake routes: (1) direct translocation across the cell membrane and (2) endocytosis after binding to cell surface heparan sulphates [6,25]. Within endocytic compartments, CPPs interact with endosomal membranes, often triggered by changes in pH in the endosomes, which results in endosome destabilization and partial cargo release [6,26]. 

Initial studies on CPP-mediated Cas9 RNP delivery used covalent attachment of polycationic CPPs to Cas9 to enhance its intracellular delivery, however, with moderate success. Low levels of Cas9-mediated insertion/deletion polymorphisms (indels) were observed at relatively high doses of Cas9-CPP conjugates [21,27,28]. Because covalent attachment might negatively influence RNP activity, either by modifying active site residues [29] or by interfering with sgRNA binding, non-covalent approaches based on electrostatic complexation of Cas9 RNP with cationic CPPs have recently been explored [22,23,30]. For stable complex formation with Cas9 RNP, cationic CPPs have been modified with one or more lipid tails [30]. It is worth emphasizing that in some of these studies these levels of efficiency were achieved by conducting experiments in conditions that are conducive to enhanced transfection rates, involving the use of Opti-MEM (a reduced serum medium). However, it is crucial to acknowledge that the effectiveness might be influenced when applying the system to cell types that are typically resistant to transfection or when working within a serum-rich environment. Compared to covalent methods, these studies have shown a remarkable increase in indel efficiencies, which could be further increased via the addition of PVA-PEG to the medium [22]. 

We previously screened a large number of different CPPs, both cationic and amphipathic, for their capacity to transfect pDNA into various cell lines [31]. From a total of >90 CPPs, only five showed robust transfection efficiencies in the presence of serum without compromising cell viability. One of these peptides, LAH5 (KKALLALALHHLAHLAHHLALALKKA), initially described by Kichler et al. (2003), showed the highest transfection efficiency of all those that were tested. This amphipathic peptide is rich in histidine residues, which are thought to be responsible for their capability to induce endosomal escape (Figure 1) [32].

In this study, the aim was to determine if LAH5 is also suitable for the delivery of Cas9 RNP to a variety of cells in the culture. The use of the LAH5 peptide for Cas9 RNP delivery or therapeutic protein delivery applications has not been explored previously. We show for the first time that LAH5 peptides form nanocomplexes with Cas9 RNP that are efficiently internalized by a variety of different cell lines. The functional delivery of Cas9 RNP is demonstrated in HEK293T cells expressing a fluorescent Cas9 reporter construct using different ratios of RNP/LAH5 nanocomplexes. Next, *CCR5* gene-targeted indels are demonstrated in various cell lines, including primary fibroblasts. Finally, we demonstrate that the co-delivery of an ssDNA HDR template with Cas9 RNP/LAH5 nanocomplexes results in efficient gene correction. To the best of our knowledge, this is the first study to show that CPPs can deliver these three different components at the same time, which are crucial for CRISPR-Cas-mediated gene correction. 

## 2. Materials and Methods

### 2.1. General Reagents

All chemicals were purchased from Sigma-Aldrich (Zwijndrecht, The Netherlands) unless otherwise specified. spCas9 and spCas9-GFP protein were purchased from Sigma-Aldrich (Zwijndrecht, The Netherlands). Modified sgRNA sequences were obtained from Sigma-Aldrich (Haverhill, UK, sequences given in Appendix A) and stored in DNAse and RNAse-free water from IDT (Integrated DNA Technologies Inc. Leuven Belgium). DNA oligonucleotides (Appendix A), HDR template (Appendix A), ATTO-550-labeled tracRNA, non-labeled crRNA (Appendix A) and Alexa-fluor-647-labeled HDR template (Appendix A) were purchased from IDT. Lipofectamine CRISPRMAX and Lipofectamine3000™ were obtained from Thermo Fisher Scientific (Waltham, MA, USA, ABD). T7 endonuclease and Q5 high-fidelity 2X master mix were purchased from New England Biolabs (Ipswich, MA, USA). The 6His-CM18-PTD4 [19], S10 [23], L17E [33] and LAH5: KKALLALALHHLAHLAHHLALALKKA peptides were purchased from Synpeptide (Shanghai, China). DNA extraction and PCR purification kits were purchased from QIAGEN Benelux B.V. (Venlo, The Netherlands). TAE buffer was purchased from Biorad (Lunteren, The Netherlands).

### 2.2. Cas9 RNP/LAH5 Nanocomplex Formation and Particle Sizing

Cas9 and sgRNA stocks were diluted in nuclease-free water (Thermo Scientific, Waltham, MA, USA). The LAH5 peptide, supplied as a powder, was dissolved in nuclease-free water with 0.1% (*v*/*v*) acetic acid at a final peptide concentration of 750 µM. sgRNA (0.75 µM) and Cas9 (0.75 µM) dissolved in HEPES buffer with a pH of 7.4 were mixed via pipetting and were incubated for 10 min at room temperature to allow the formation of the Cas9 ribonucleoprotein (RNP) complex. Subsequently, an equal volume of LAH5 (37.5 µM to 187.5 µM) was added to the RNP and incubated for 10 min at RT to enable nanocomplex formation. The obtained molar ratios of Cas9:sgRNA:LAH5 varied between 1:1:50 and 1:1:250. Following nanocomplex formation, samples were diluted in HEPES buffer with a pH of 7.4; then, 100 µL of the sample was transferred into a lowest sizing cell cuvette (Malvern, Malvern, UK), and size distribution was measured with a Malvern Zetasizer Nano-ZS (Malvern Instruments, Worcestershire, UK). Subsequently, formulated nanocomplexes were diluted 8× times in 10 mM HEPES buffer with a pH of 7.4; then, the zeta potential was measured with Zetasizer Nano Z (Malvern ALV CGS-3, Worcestershire, UK). All measurements were performed in triplicate. 

### 2.3. Electrophoretic Mobility Shift Assay (EMSA)

Peptide-mediated complexation was determined by performing a fluorescence-based electrophoretic mobility shift assay by using a non-stained and non-denaturing 1.5% agarose gel in TAE buffer with a pH of 8.3. For this, Cas9-GFP (Merck), ATTO-550-labeled tracrRNA, unlabeled crRNA (IDT), Alexa-647-labeled HDR template (IDT) and unlabeled LAH5 peptides were used. Labeled components and unlabeled LAH5 peptides were formulated at various molar ratios as described above. All labeled components were diluted such that the final concentration reached 1 µM when loaded on gel. All formulated nanocomplexes were adjusted to a total volume of 10 µL by adding nuclease-free water. Then, 2 µL of 40% Glycerol was added, after which samples were loaded into the wells of the gel. The agarose gel was run at 170 V for 15 min. Gel images were captured by ChemiDoc^TM^ XRS+ (Bio-Rad, Lunteren, The Netherlands), and fluorescent images were merged and analyzed with ChemiDoc^TM^ XRS+ (Bio-Rad) with Image lab software version 5.0. 

### 2.4. Cell Lines and Cell Culture

HEK293T (CRL-3216), HeLa (CCL-2), HEPG2 (HB-8065) and primary fibroblast cells (CCL-186) were all obtained from the American Type Culture Collection (ATCC, Manassas, VA, USA) through LGC Standards (Molsheim, France). The HEK293T stoplight cell line was constructed as described previously [34]. HEK293T stoplight cells, HEK293T HDR stoplight cells, HEK293T and primary fibroblast cells were cultured in a high-glucose DMEM medium, and HeLa and HEPG2 cells were cultured in Eagle’s minimum essential medium. The culture medium was supplemented with 10% (*v*/*v*) FBS (Sigma Zwijndrecht, The Netherlands), and cells were cultured at 37 °C and 5% CO_2_. Cell culture plastics, unless otherwise specified, were purchased from Greiner Bio-One (Alphen aan de Rijn, The Netherlands).

### 2.5. Generation of HEK293T HDR Stoplight Reporter Cell Line

The HDR fluorescent stoplight reporter construct was synthesized by Integrated DNA Technologies (Leuven, Belgium). The reporter construct was cloned into a pHAGE2-MCS-EF1a-IRES-NeoR plasmid using NotI and BamHI restriction enzymes and a NEB Quick Ligation Kit (all from New England Biolabs, MA, USA). The pHAGE2-EF1a-HDR stoplight-IRES-NeoR plasmid was then used for lentiviral production and transduction. HEK293T cells were transfected with pHAGE2-EF1a-HDR Stoplight-IRES-NeoR, MD2.G plasmid and PSPAX2 plasmid (Addgene #12259 and #12260, respectively) at a 2:1:1 ratio using 3 µg of 25 kDa linearized PEI per µg of DNA. The transfection medium was removed after 24 h and replaced with DMEM with 10% FBS. After 48 h, the lentiviral supernatant was harvested, and cells were cleared via a 5 min 500× *g* centrifugation step, followed by 0.45 µm syringe filter filtration. HEK293T cells were transduced with lentiviral stocks overnight with 8 µg/µL polybrene (Sigma, Zwijndrecht, The Netherlands). After 24 h, the transduction medium was removed and replaced with a culture medium supplemented with 1500 µg/mL G418. After 5 days, the concentration of G418 was lowered to a maintenance concentration of 1000 µg/mL G418. After two weeks of culturing in the presence of selection antibiotics, the reporter cells were sorted for mCherry^+^eGFP^−^ fluorescent signals on a BD FACSAria III cell sorter.

### 2.6. Transfection Experiments

All cell types used in this study were plated at 30,000 cells per well in a 96-well plate (Greiner M0687-100EA) and were incubated for 24 h at 37 °C and 5% CO_2_ to reach a confluency of 50%. Next, cells were transfected with Cas-CPP or Cas-HDR-CPP. As a positive control for transfection, spCas9 RNP complexed with CRISPRMAX was used according to the manufacturer’s instructions (Thermo Fisher Scientific). Cas-CPP nanocomplexes were prepared following RNP formation. Increasing concentrations of LAH5 peptide were mixed with RNP by keeping the ratio of RNP:LAH5 between 1:50 and 1:250. Then, Cas-CPP mixtures were incubated for 10 min to form nanocomplexes. For HDR experiments after the preparation of RNPs, various concentrations of HDR template were added to the preformed spCas9 RNP to obtain multiple ratios of 1:1, 1:2 and 1:4, and they were incubated for 5 min at room temperature. Finally, LAH5 peptide was added to the spCas9 RNP and HDR template mixtures and incubated for 10 min to form Cas-HDR-CPP nanocomplexes. Opti-MEM was added to reach a total volume of 160 μL before removing culture medium and adding the nanocomplexes to the cells at the indicated concentrations. Cas9, sgRNA and HDR template concentrations were 20 nM at a 1:1:1 molar ratio in the 96-well plate. A duration of 24 h after transfection, cells were washed 2 times with fresh media. Cells were incubated for another 24 h at 37 °C and 5% CO_2_. In case DNA extraction was needed for genetic analysis (T7 endonuclease and TIDE), the transfection experiments were performed in 48-well plates (Greiner M8937-100EA) to ensure sufficient amounts of DNA for post-analysis.

### 2.7. Stoplight Gene Editing and Correction Assays

To assess the efficiency of delivery at the cellular level, upon transfection with Cas-CPP nanocomplexes, two different reporter cell lines, HEK293T NHEJ stoplight cells [34] and HEK293T HDR stoplight cells, were used. A duration of 48 h after transfection, gene editing and gene correction efficiencies were assessed using flow cytometry analysis. 

### 2.8. Flow Cytometry to Determine Gene Editing and Gene Correction Efficiencies in HEK293T Stoplight and HEK 293T HDR Stoplight Cells

Cells were harvested by washing twice with PBS, followed by a trypsinization step, after which cells were fixed in 1% paraformaldehyde. Cells were washed in PBS and transferred to a BD Falcon U-bottom 96-well plate (Becton Dickinson, Franklin Lakes, NJ, USA) for flow cytometry analysis. Reporter fluorescence was detected via flow cytometry using BD FACS CANTO II (Becton Dickinson, Franklin Lakes, NJ, USA). mCherry was measured using the PerCP-Cy5-5-A channel of the flow cytometer, and eGFP fluorescence was determined in the FITC channel. The flow cytometry results were analyzed with Flowlogic software (Inivai Technologies, Mentone, Australia, version 8.7). The gating strategy used for the flow cytometry analyses, both for the HEK293T stoplight and HEK293T HDR stoplight cells, is specified in Appendix A. Both gene editing and gene correction efficiency, depending on the cell line, were identified as the number of mCherry^+^ cells expressing eGFP, as described previously [34].

### 2.9. Confocal Microscopy 

HEK293T stoplight cells were plated at a quantity of 30,000 cells per well in a 96-well imaging plate (Greiner CellStar #655090). After 24 h of incubation, cells were treated with different concentrations of Cas-CPP and Cas-HDR-CPP nanocomplexes. After 24 h of incubation at 37 °C and 5% CO_2_, cells were washed with 100 μL of high-glucose DMEM medium supplemented with 10% FBS. Following this, cells were incubated for another 24 h at 37 °C and 5% CO_2_. Cell nuclei were stained by adding 2 µg/mL Hoechst 33342 in a complete cell culture medium as the final concentration and incubated for 30 min, after which cells were imaged using the Yokogawa CV7000 Confocal Microscope. (Yokogawa Corporation, Tokyo, Japan).

### 2.10. Cytotoxicity Assays

The cytotoxicity of the RNP/LAH5 peptide nanocomplexes was evaluated via an MTS (cell viability) assay [35]. Cells were seeded into 96-well plates with 50% confluency and were incubated for 24 h at 37 °C and 5% CO_2_. Cas-CPP nanocomplexes (prepared at molar ratios of 1:50 to 1:1000) were added at a concentration of 20 nM of RNP per well in a total volume of 100 μL of Opti-MEM. A duration of 24 h post-transfection, cytotoxicity was determined with the CellTiter 96^®^ AQueous One Solution Cell Proliferation Assay (MTS) (Promega, Madison, WI, USA) according to the manufacturer’s protocol. The absorbance was measured at 490 nm on a Bio-Rad iMark microplate reader model 1681130. The relative cell viability was calculated by setting the absorbance value of untreated cells to 100%, and the absorbance value for those treated with 1% Triton X-100 was set to 0%.

### 2.11. Cell Uptake Assay

Cas9-GFP (Sigma), ATTO 550 tracrRNA (IDT) and crRNA (IDT), specific for the *HPRT* housekeeping gene and Alexa-647-labeled single-stranded HDR template, were used to prepare fluorescent Cas-CPP nanocomplexes for cellular uptake experiments. First, ATTO 550 tracrRNA (IDT) and crRNA (IDT) were mixed at a 1:1 ratio to form an ATTO-550-labeled sgRNA complex. Then, the uptake of these components was tested with and without complexation with Cas9-GFP (10 nM) and ATTO 550 sgRNA (10 nM) (RNP) with 3 μM LAH5 peptide at a 1:150 ratio (m/m). A complex consisting of 10 nM Cas9-GFP, ATTO 550 sgRNA (RNP) and Alexa 647 HDR template at equimolar concentrations was prepared following the previously described procedure. Additionally, the labeled RNP and labeled HDR template were mixed with 3 µM LAH5 peptide at a ratio of 1:1:150 (m/m). Following the preparation of the nanocomplexes, HeLa cells were transfected in a 96-well black plate. A duration of 24 h after transfection, the nuclei of HeLa cells were stained by supplementing the complete cell medium with Hoechst 33342 dye at a final concentration of 2 µg/mL, followed by incubation for 30 min. Microscopy images were recorded at 60× magnification using the Yokogawa CV7000s confocal microscope. In addition, HeLa cells were transfected with a CD63-eGFP plasmid using Lipofectamine 3000 according to the manufacturer’s protocol. A duration of 48 h after plasmid transfection, cells were treated with Cas9/ATTO 550 sgRNA(RNP) with and without LAH5 complexation, as described above.

### 2.12. T7 Endonuclease Assay 

A T7E1 assay was performed to detect the insertion/deletion (indel) frequency after gene editing [36]. Genomic DNA was extracted from the cells 48 h after transfection with LAH5 peptide/RNP nanocomplexes using the Qiagen DNeasy Blood & Tissue Kit (Qiagen GmbH, Hilden, Germany) following the manufacturer’s instructions. PCR was performed using the primers designed for the sgRNA target locus (Appendix A) using Q5^®^ Hot Start High-Fidelity 2X Master Mix (New England Biolabs, MA, USA). Afterward, PCR products were purified using the QIAquick PCR Purification kit (Qiagen GmbH, Hilden, Germany). PCR products were denatured at 95 °C for 10 min in the presence of NEBuffer 2 (New England Biolabs), and they were annealed by slowly lowering the temperature (95–85 °C at 2 °C per second and 85–25 °C at 1 °C per second). Subsequently, re-annealed PCR products were incubated with 5U T7E1 enzyme (New England Biolabs, MA, USA) at 37 °C for 18 min to cut heteroduplexes. DNA products were run on a 2% agarose gel in TAE buffer with a pH of 8.3 (Biorad). The indel frequency was calculated by determining the intensities of cleaved and uncleaved bands based on a densitometry analysis using ImageJ version 1.53t. 

### 2.13. TIDE Analysis (Tracking of Indels via Decomposition)

Genomic DNA was isolated, and the sgRNA target genomic region was amplified via PCR using the same methods as described in the section on the T7E1 assay (primers are listed in Appendix A). The PCR products were purified using the QIAquick PCR Purification kit (Qiagen GmbH, Hilden, Germany) and were submitted for unidirectional Sanger sequencing (Macrogen Europe). Afterward, Sanger sequence chromatograms of purified PCR products were used for a TIDE analysis (http://tide.nki.nl, accessed on 6 November 2022) [37]. Gene modification frequencies were determined using the sequencing chromatogram from negative control cells as a reference and by comparing the sequence chromatogram from treated samples. During the analysis, parameters were set to detect a maximum indel size of 15 nucleotides. The decomposition frame was set to the default parameters [37].

### 2.14. Statistical Analysis

Statistical analysis was performed using GraphPad Prism (9.4.1). The statistical analysis methods that were used are specified under the specific figure legends. Values in all experiments are represented as means ± standard errors of the mean (SDs) of at least three independent experiments performed in duplicate. An increase in the delivery efficiency was considered significant at * *p* < 0.05 using an analysis of variance (ANOVA) with Dunnett’s multiple comparisons test or an ANOVA with Bonferroni’s multiple comparisons test.

## 3. Results

### 3.1. Complexation of Cas9 RNP and HDR Template with LAH5 Peptides

Because we previously observed that LAH5 peptide shows the robust transfection capabilities of pDNA in high-serum conditions [31], we decided to investigate its potential for direct Cas9/RNP delivery. To this end, we conducted an initial comparison of the delivery efficiency of Cas9/RNP using previously documented peptides, namely 6His-CM18-PTD4 [19], S10 [23] and L17E [33], alongside LAH5 peptide (Appendix A). In this initial screening, LAH5 outperformed the other peptides in facilitating direct Cas9/RNP delivery to HEK293T cells, leading us to further characterize and explore its potential for Cas9-mediated gene editing.

LAH5, having a net positive charge at a neutral pH, is known to form stable nanocomplexes with pDNA due to electrostatic complexation [32]. Because the predicted isoelectric point of spCas9 is 9 (ExPASy) and is thus positively charged at a physiological pH, electrostatic complexation might not be obvious. Therefore, we first investigated the interaction of LAH5 peptide with the spCas9 protein with and without associated sgRNA and in the presence or absence of ssDNA (HDR template) using an electrophoretic mobility shift assay (EMSA). For this assay, fluorescently labeled spCas9 (spCas9-GFP), tracrRNA (ATTO 550) and HDR template (Alexa 647) were used. Increasing molar ratios of LAH5 to Cas9-GFP protein were tested for complex formation (Figure 2). As expected, the spCas9-GFP protein (pI = 8.8) itself had poor electrophoretic mobility in the agarose gel due to the presence of few negative charges under the tested conditions (TAE buffer, pH of 8.3). When complexed with sgRNA, the electrophoretic mobility of spCas9-GFP increased, showing a band on gel with both GFP and ATTO 550 fluorescence in addition to free sgRNA. At molar ratios of LAH5:Cas9-GFP RNP > 50, the mobility of the Cas9-GFP RNP was lost, which is indicative of LAH5-mediated complex formation (Figure 2A). A similar trend was seen in the presence of HDR template, which could be readily complexed together with Cas9-GFP RNP by LAH5 peptide at molar ratios >50 (Figure 2B). Overall, LAH5 peptides at >50 times the molar excess formed supramolecular complexes with RNP alone and with RNP and HDR template together. 

### 3.2. Nanocomplex Size and Zeta Potential

The size distribution and the zeta potential of RNP/LAH5 nanocomplexes prepared at molar ratios of 1:50 to 1:250 were determined (Figure 3). LAH5 peptide, when mixed with Cas9 RNP, formed nanocomplexes with an average size ranging from 200 to 400 nm, depending on the ratios of LAH5 peptide to Cas9 RNP (Figure 3A). Using ratios above 1:200 resulted in the formation of substantially smaller particles compared to the lower ratios that were tested. This can be explained by the relatively weak electrostatic interactions between LAH5 peptide monomers and Cas9 RNP, generating highly dynamic interpolyelectrolyte complexes from which LAH5 peptides continuously associate and dissociate. Having more LAH5 peptides in the solution shifts the equilibrium to more LAH5 peptides being bound, thereby shielding potentially exposed negatively charged surfaces on the interpolyelectrolyte complexes that could cause aggregation into larger structures. The relatively high polydispersity index of the nanocomplexes ranged from 0.3–0.4, which shows that the formed complexes were not uniform in size, also pointing to the highly dynamic nature of these RNP/LAH5 nanocomplexes. Zeta potential measurements of the nanocomplexes showed an increase in the zeta potential with increasing concentrations of LAH5 peptide, but in all cases, nanocomplexes were negatively charged. When nanocomplexes were formed by mixing equimolar quantities of sgRNA and Cas9 and by increasing the concentration of LAH5 peptide, we observed an increased zeta potential shift. Specifically, the zeta potential values shifted from around -8 mV to approximately −4 mV when using 50× and 250× peptide concentrations, as illustrated in Figure 3B. This is primarily attributable to the cationic characteristics inherent to LAH5 peptide.

### 3.3. Cellular Uptake and Functional Gene Editing in HeLa Cells

The cellular uptake of fluorescently labeled Cas9 RNP (Cas9-GFP with ATTO-550-labelled sgRNA) in a free form or complexed with LAH5 peptide at a 1:150 molar ratio was monitored via live cell confocal imaging in HeLa cells (Figure 4A). A duration of 24 h post-transfection, the cellular uptake of Cas9 RNP without peptide complexion exhibited negligible levels, with no discernible GFP signal spots or ATTO 550 signals evident within HeLa cells. This observation may imply that, without peptide complexation, the presence of fluorescently labeled components within the cytosol of the cells is undetectable. When Cas9 RNP was complexed with LAH5 peptide, a large increase in cell-associated fluorescence was observed, with both Cas9-GFP and ATTO-550-sgRNA co-localizing inside the cells.

Furthermore, the Cas9-sgRNA ATTO 550 (RNP)/LAH5 nanocomplex showed co-localization with CD63-eGFP, an eGFP-tagged late endosome marker, indicating that the internalization of RNP/LAH5 nanocomplexes occurs via the endocytic pathway. Moreover, a punctated pattern of intracellular fluorescence in line with endocytic uptake can be seen, although the possibility of limited direct membrane translocation cannot be entirely ruled out (Appendix A). To demonstrate the functional delivery of Cas9 RNP, which requires nuclear delivery of the RNP and hence endosomal escape, we measured the level of gene editing in HeLa cells. For this, an sgRNA that targets the human *CCR5* gene was used. Cas9 RNP in a free form or complexed with 150-fold molar access of LAH5 peptide (3 μM) was added to the cells. A duration of 48 h following transfection, genomic DNA was isolated, and the target region of the *CCR5* gene was amplified via PCR. Gene editing was assessed semi-quantitatively via a T7 endonuclease assay (Figure 4C) and quantitatively via the TIDE analysis assay (Figure 4D) [37]. Both assays showed no or low levels (<1%) of gene editing with Cas9 RNP in a free form. However, when complexed with LAH5 peptide, gene editing could be clearly observed, reaching ~17% editing as determined by the TIDE assay (Figure 4D). 

Regarding precise gene correction, an HDR template is required to guide homology-directed repair following a CRISPR-mediated DSB. We next tested the possibility to co-deliver an ssDNA HDR template of 81 bp with the Cas9 RNP/LAH5 nanocomplexes (Figure 4B). For this, an Alexa-647-labeled HDR template and Cas9 RNP at a 1:1 molar ratio were complexed with 250-fold molar access of LAH5 peptide. Upon delivery, it can be seen that the HDR template co-localized with the Cas9 RNP inside HeLa cells, suggesting that the HDR template was incorporated into the LAH5 nanocomplexes, resulting in intracellular co-delivery with the Cas9 RNP complexes. No cellular uptake of the HDR template was observed when mixed with Cas9 RNP without LAH5 peptide. 

### 3.4. Cytotoxicity Study

The cytotoxicity of RNP/LAH5 nanocomplexes in multiple cell lines was determined by performing an MTS assay (Appendix A). No severe signs of toxicity were observed in any of the tested cell lines after exposure to the 1:50 to 1:500 ratio complexes, demonstrating that the RNP/LAH5 complexes were not toxic to the treated cell types at the concentrations that were used for transfection. However, when the peptide concentration increased to a ratio of 1:1000, severe toxicity was observed in all tested cell lines, resulting in up to a 50% decline in cell viability. The results suggest that RNP/LAH5 nanocomplexes are not toxic at ratios that are normally used for transfection (in the range of 1:50 to 1:250).

### 3.5. Gene Editing Efficiency

To further evaluate the functional delivery of Cas9 RNP using LAH5, a stoplight reporter cell for Cas9 activity (Figure 5A) was used [34]. These HEK293T stoplight reporter cells constitutively express mCherry (a red fluorescent protein). eGFP (green fluorescent protein) expression can be induced upon CRISPR-Cas9-mediated targeted disruption of a target sequence downstream of the mCherry sequence, which results in a +1 or +2 frameshift to bypass a stop codon and bring the eGFP in frame with mCherry. This allows quick assessment of the indel efficiency and thereby the detection of the most optimal molar ratio between Cas9/RNP and LAH5 peptide. To assess the delivery efficiency, HEK293T stoplight cells were incubated with various combinations of Cas9 RNP/LAH5 nanocomplexes for 24 h in Opti-MEM medium. A duration of 24 h after transfection, the medium was replaced with fresh DMEM supplemented with 10% FBS, and cells were incubated for an additional 24 h. Cellular fluorescence was determined via both flow cytometry and confocal microscopy. The flow cytometry analysis showed that increasing the amount of peptide at a fixed Cas9 RNP concentration (20 nM) resulted in increasing gene editing. Around 70% of cells were successfully edited at the highest LAH5 peptide concentration tested (1:250 molar ratio of Cas9 RNP:LAH5 peptide) (Figure 5B,C). T7E1 and TIDE assays were used to corroborate these findings at the DNA level (Figure 5E,F). In line with the flow cytometry results, these assays showed increased levels of gene editing with increasing ratios of LAH5 to Cas9 RNP. 

We next varied the amount of Cas9 RNP (5–40 nM) at a fixed LAH5 peptide concentration of 5 µM (Figure 5F,G). As expected, the lowest concentration of Cas9 RNP resulted in the lowest gene editing efficiency. Increasing the Cas9 RNP dose did not result in a linear increase in editing efficiencies, but it seemed to reach a saturation point at approx. 70% of gene editing, which is close to the theoretical maximum frameshift frequency of +1 nt and +2 nt of 80% according to an inDelphi in situ prediction for this reporter assay [34]. No gene editing was observed when Cas9 RNP was prepared with a control sgRNA (non-target sgRNA, Appendix A) to confirm sequence specificity (Figure 5C). Gene editing was confirmed at the genomic level via the T7E1 assay (Figure 5G). Taken together, these results demonstrate that optimal gene editing can be achieved at a Cas9 RNP concentration of 20 nM complexed with 5 μM LAH5 peptide. At these concentrations, cell viability was >80% compared to control conditions, and the maximum theoretical activation of the fluorescent stoplight reporter was approached [34] (Figure 5F,G and Appendix A). 

Furthermore, to obtain insight into the kinetics of the delivery of Cas9 RNP by LAH5, we measured the effect of the incubation time of reporter cells with the LAH5:Cas9 RNP complexes on the overall editing efficiencies. In short, HEK293T stoplight cells were incubated from 1 min up to 24 h with the Cas9/LAH5 nanocomplexes, after which cells were washed three times with PBS, and a fresh medium was added. Cells were cultured for a total of 48 h after transfection before flow cytometry analysis for eGFP expression (Appendix A). The results demonstrate that LAH5-mediated transfection already reached 60% of gene editing when cells were exposed for only 3 h to the nanocomplexes, and similar transfection efficiencies were seen after 24 h of incubation with the lipid-based CRISPRMAX transfection agent. LAH5 strongly outperformed CRISPRMAX for the 1 and 3 h incubation times. 

This suggests that Cas9 RNP reaches the cytosol much faster when delivered with LAH5 peptides than when complexed with a lipid-based transfection agent. This is most likely due to the pH-dependent membrane destabilizing effect of LAH5, which can trigger the endosomal escape of its cargo [32].

The indel efficiencies that were determined via flow cytometry were consistently higher than those determined via the TIDE assay (Figure 5C,D). The higher values obtained with flow cytometry could be explained by a high copy number of the reporter construct that was introduced via lentiviral transduction. If a reporter cell line has multiple copies incorporated in its genome, editing one locus leads to a GFP-positive signal with flow cytometry. However, at the same time a high copy number also results in an underestimation of indels using TIDE analysis. Nevertheless, both the flow cytometry and TIDE results confirm that the optimal ratio of the RNP/LAH5 nanocomplex, in terms of the efficiency of gene editing, is 1:250.

### 3.6. CCR5 Editing Efficiency in Different Cell Types

We tested gene editing efficiency with Cas9/LAH5 nanocomplexes in several different human cell types: HEK293T, HeLa, HEPG2, ARPE-19 and primary human fibroblasts, targeting the human *CCR5* gene. The gene editing outcome was determined via T7E1 and TIDE assays. The sequencing data of both the negative controls and treated groups were analyzed via TIDE. Cells were treated with Cas9/LAH5 nanocomplexes, and different ratios of Cas9/LAH5 were tested, ranging from 1:50 to 1:250, respectively. As a positive control, CRISPRMAX was used (Figure 6 and Appendix A). The editing efficiencies between the tested cell types varied substantially, with primary human fibroblasts showing the lowest level of gene editing and HeLa cells showing the highest. Despite these differences, all cell types showed a similar dose response, with higher amounts of LAH5 peptide leading to higher editing efficiencies. These results demonstrate the functionality and effective delivery potential of LAH5 in vitro.

### 3.7. HDR-Dependent Gene Correction

A newly developed HEK293T HDR stoplight reporter system was used as a read-out to evaluate LAH5-mediated HDR gene correction efficiency. HEK293T HDR stoplight cells constitutively express mCherry, directly followed by a stop codon and an eGFP open reading frame (ORF). Including a single-stranded oligonucleotide (ssODN) HDR template with CRISPR-Cas9 induces a specific replacement of the stop codon with a glutamine, resulting in the expression of an mCherry–eGFP fusion protein (Figure 7A). HEK293T HDR stoplight cells were incubated with fixed amounts of Cas9 RNP (20 nM) and HDR ssDNA template (20 nM), which were complexed with increasing amounts of LAH5 peptides (50- to 250-fold molar excess compared to RNP) for 24 h. A duration of 48 h after transfection, HEK293T HDR stoplight cells were tested for mCherry and eGFP fluorescence via flow cytometry (Figure 7 and Appendix A). We observed that increasing amounts of LAH5 led to increasing gene correction efficiencies (Figure 7C). The nanocomplex that triggered the highest gene correction efficacy was Cas9/HDR template/LAH5 peptide nanocomplexed at a 1:1:250 molar ratio. This ratio generated more than 20% gene correction in the total cell population. The efficiency of HDR when the cells were treated with nanocomplexes prepared at RNP/HDR (1:2) molar ratios and complexed with LAH5 peptide concentrations at 1 μM to 5 μM is given in Appendix A. The level of gene correction was consistently lower compared to nanocomplexes containing Cas9 and HDR template at a 1:1 molar ratio. 

Next, we tested the effect of the total concentration of Cas9 RNP and HDR template on gene correction efficiency in HEK293T HDR reporter cells. For this, the concentrations of Cas9 RNP to HDR template were varied between 10 and 40 nM and at molar ratios of 1:1, 1:2 and 1:4. The results show that gene correction increased between the 10 nM and 20 nM Cas9/HDR template at a 1:1 molar ratio, but higher concentrations did not further increase gene correction efficiencies (Figure 7C). We also tested the impact of increased amounts of HDR template (1:2 and 1:4 molar ratios). Although the gene correction % was stable in cells treated with nanocomplexes containing RNP (20 nM)/HDR template (20 nM) to RNP (40 nM)/HDR template (40 nM) at a 1:1 ratio, increased ratios of 1:2 to 1:4 RNP/HDR template caused a decreasing trend of HDR (Appendix A). These data demonstrate that nanocomplexes prepared at a stable concentration of LAH5 peptide reached a gene correction efficacy of around 20%; however, increased concentrations of RNP and HDR template did not further increase gene correction. Moreover, further increasing the ratio of the HDR template impacted gene correction adversely.

Based on the statistical analysis, the populations of the cells treated with RNP/HDR template/LAH5-mediated nanocomplexes in various conditions had significantly higher gene correction outcomes compared to the non-treated negative controls. The efficiency was dependent on increasing ratios of LAH5, as illustrated in Figure 7D. The HDR efficiency reached around more than 20% at 1:1:200 and 1:1:250 ratios of RNP/HDR template/LAH5 peptide (Figure 7D). 

## 4. Discussion

For efficient genome editing, the intracellular delivery of CRISPR-Cas9 components is essential. This report investigates a promising delivery strategy, wherein CPPs are employed as the chosen vehicle for effective delivery. Peptide-based delivery offers an advantage in which pre-assembled Cas9 RNP can be complexed and delivered and, depending on the peptide carrier used, induce endosomal escape for the cytosolic delivery of the complexes. In this report, we repurposed the LAH5 amphipathic peptide, initially developed for the transfection of pDNA [32] for CRISPR-Cas9 RNP delivery to a variety of human cells in vitro. We demonstrated that LAH5 peptide can form stable nanocomplexes with Cas9 RNP and with Cas9 RNP combined with an ssDNA template for homology-directed repair. We are the first to demonstrate that Cas9 RNP and an ssDNA template can be stably co-entrapped inside LAH5 peptide nanocomplexes. The electrophoretic mobility shift assay (EMSA) demonstrates that, at a molar excess >50 for LAH5:Cas9 RNP, stable nanocomplexes can be formed via electrostatic complexation. Even though hydrophobic clustering of LAH5 peptide itself cannot be excluded, as was demonstrated before [38], and the formed nanocomplexes are polydisperse, we observed stable sizes of the nanocomplexes at 1:200 and 1:250 RNP/LAH5 ratios and confirmed this interaction via an EMSA assay. Moreover, elevating the concentration of the LAH5 peptide in the RNP/peptide nanocomplex results in the formation of smaller complexes. This reduction in size can be attributed to the shift in the equilibrium between LAH5 peptides that are free in the solution and LAH5 peptides that are complexed with RNPs. As a result, potentially exposed negative surfaces are better shielded by cationic LAH5 peptides, leading to nanocomplexes that are less prone to aggregating into larger structures. It is also worth noting that the protein and nucleotide labeling strategies that are employed to visualize these molecules within EMSA assays may have some effects on particle complexation, which is an inherent challenge in such assays. However, we believe that these effects are limited, as Cas9 GFP tagging barely affects its isoelectric point—the main determinant for peptide nanocomplexation—shifting from 9.0 to 8.8, according to ExPASy isoelectric point analysis. Moreover, ATTO labels are commonly employed to label sgRNAs while retaining their functionality, suggesting that, after this modification, the secondary structure (and thus its complexation with Cas9) remains intact. It is important to note that further biophysical and structural characterization of LAH5 nanocomplexes is needed to elucidate the stoichiometry of each of the components in the nanocomplexes, as well as the topology of the supramolecular multicomponent assemblies. It is at present unclear what the dynamics of release are for each of the cargoes. It is also unclear if LAH5 is accessible on the surface of the nanocomplexes for membrane interaction or if it needs to be released first in order to be membrane-active. Nevertheless, because we demonstrated that these multiple components can be complexed by LAH5, this approach offers the opportunity to efficiently deliver multiple components that are needed for gene editing with a single delivery system. To show a proof of concept for this, we demonstrated that the co-delivery of an ssDNA template with Cas9 RNP led to targeted gene correction in a newly developed reporter cell line. To our knowledge, this is the first time that HDR-mediated targeted gene correction via the delivery of all components with a peptide-based delivery system was demonstrated. A recent publication by Foss et al. elegantly showed the CPP-mediated delivery of CRISPR-Cas9 RNP into human lymphocytes for gene knock-out, but it relied on the co-delivery of an AAV vector carrying the HDR template for the targeted insertion of a gene construct encoding chimeric antigen receptors [39]. However, the study did not demonstrate peptide-mediated RNP/additional delivery of DNA templates for targeted HDR-mediated gene repair. In another recent report, an amphiphilic peptide-mediated delivery system was used for the delivery of RNPs without an additional DNA template to edit primary human lymphocytes [40]. In the context of our study, it is noteworthy to emphasize the uniqueness of our findings, as we successfully demonstrated the exclusive ability of a peptide-based system to co-deliver both RNP and HDR templates. This distinctive aspect sets our study apart from the existing literature in the field.

It is crucial to acknowledge that the high editing efficiencies observed in this study may not necessarily translate to scenarios requiring the co-delivery of larger DNA fragments to facilitate HDR-directed repair or correction in non-dividing cells. A notable limitation of synthetic delivery systems employing cell-penetrating peptides (CPPs) likely lies in their inability to actively facilitate nuclear trafficking, despite their effectiveness in intracellular delivery. Consequently, further investigation is needed to assess the suitability of CPP-based delivery strategies for HDR in non-dividing or slow-dividing cells, similar to polyplexes and lipoplexes [41,42].

When comparing our results with previous publications on CPP-assisted CRISPR-Cas delivery, it is important to note that our LAH5-based transfection was active at much lower concentrations of Cas9 RNP compared to several recently published peptide-mediated delivery studies [39,40]. In fact, the optimal concentrations at which we observed editing (20 nM of Cas9) are 10–1000-fold lower than these published results [39,40]. This observation may be attributed to the high histidine content that was present in our cell-penetrating peptide (CPP), which possesses a more favorable pKa for effective endosomal escape compared to peptides with lower histidine residue compositions in their sequences [43]. This has obvious advantages, as high concentrations of Cas9 RNP may cause innate immune responses by engaging pattern recognition receptors such as TLR3, RIG-I and PKR, which in turn may lead to Cas9-specific adaptive responses [44,45]. Being able to lower the dose may reduce such undesired immune effects. Interestingly, despite a negative zeta potential at a pH of 7.4, we observed fast uptake and delivery of our LAH5-RNP complexes, demonstrating efficient cargo delivery after cell exposure of only 3 h (Appendix A). Whereas a negative charge of nanoparticles generally leads to decreased endosomal uptake, other factors such as charge density and hydrophobicity are of equal importance and may explain these results [46]. Work by Kichler et al. showed through the use of (ATR)-FTIR spectroscopy that the LAH4 family exhibits alpha-helical confirmations with all positive histidine residues located on one side of the helix [32]. Further NMR spectroscopy indicated that the interactions of these CPPs with membranes were modulated by these histidine-rich side-chains. Moreover, at a pH level of 5.4, the nanocomplexes underwent a shift toward a positive charge (Figure 3C). This characteristic empowers the RNP: LAH5 nanocomplexes to proficiently interfere with endosomal membranes, capitalizing on the pH disparity. It is worth emphasizing that the pH level experiences fluctuations during the process of endosomal uptake, and the acidic milieu within the endosome plays a pivotal role in facilitating this phenomenon.

Furthermore, the robustness of our peptide-based transfection was demonstrated by showing gene editing in a variety of human cell types. Even though the editing efficiencies varied from 20 to 70%, in all cases, sufficiently high levels of editing were achieved to be therapeutically relevant. LAH5 peptide demonstrated significantly superior performance compared to CRISPRMAX at shorter incubation times. This observation strongly suggests that the cytosolic delivery of Cas9 RNP occurs more rapidly when facilitated by LAH5 peptide as compared to complexation with a lipid-based transfection agent.

This work demonstrates that the amphipathic LAH5 peptide can efficiently deliver Cas9 RNP with and without HDR template in a variety of cell types by means of nanocomplex formation and concomitant intracellular delivery of its cargo. The editing efficiencies were unprecedented, showing high levels of gene editing at Cas9 RNP concentrations as low as 5–10 nM. 

We do not expect that the current nanocomplex formulation is stable enough for direct in vivo applications. To make nanocomplexes, a large molar excess of LAH5 peptide was needed. This points to a weak interaction with Cas9 RNP. Intravenous injection would quickly dilute out the free peptide, leading to a change in the equilibrium and potential dissociation of the nanocomplexes. Nevertheless, future research will investigate nanocomplexes’ stability under simulated in vivo conditions (whole serum) to obtain more insights into the stability of these complexes. For now, LAH5-peptide-assisted delivery of Cas9 RNP could be very useful, e.g., for the ex vivo modification of NK or T cells, as it relies on a single peptide that can be easily synthesized under good manufacturing practice (GMP).

## Figures and Tables

**Figure 1 pharmaceutics-15-02500-f001:**
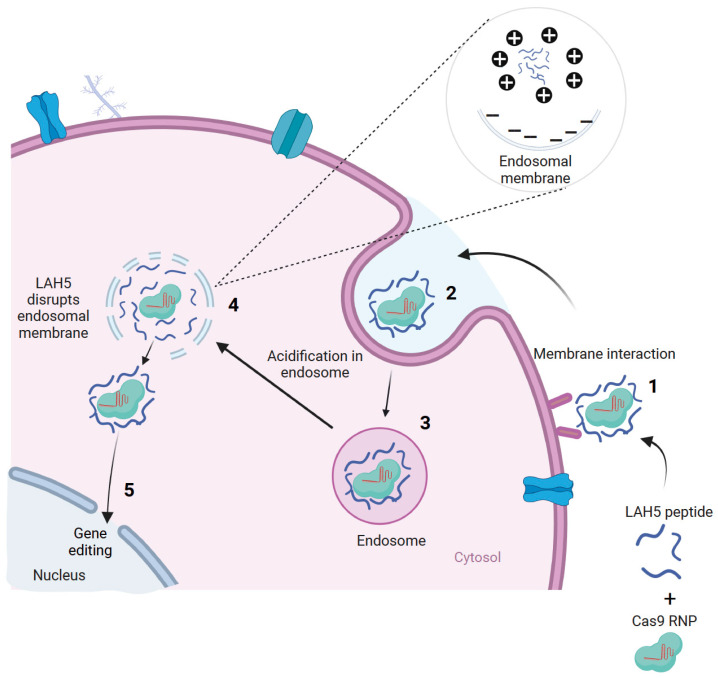
Schematic displaying how Cas9 RNP/CPP complexes are hypothesized to be taken up and subsequentially escape from the endosomal pathway. First, the nanocomplex interacts with the cellular membrane of target cells (1) and is subsequently internalized (2) via endocytosis (3). Following acidification of the endosome, the amphipathic cell-penetrating peptides induce lytic membrane interactions with the endosomal membrane. Following endosomal membrane disruption, nanocomplexes are released into the cytosol (4). Nuclear localization signal (NLS) sequences on Cas9 allow internalization into the nucleus, where gene editing can occur (5). Created with BioRender.com, accessed on 24 December 2022.

**Figure 2 pharmaceutics-15-02500-f002:**
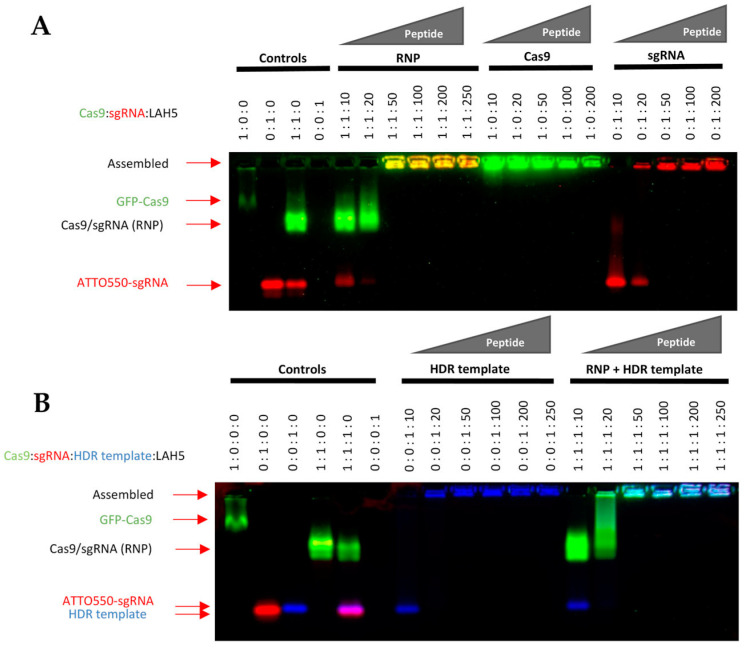
Electrophoretic mobility shift assay (EMSA) of Cas9-GFP, Cas9-GFP RNP (**A**) and HDR DNA template alone or combined with Cas9-GFP RNP. Cas9 fluorescence signal is shown in green, sgRNA fluorescence signal is shown in red. (**B**) sgRNA was labeled with ATTO 550, and the HDR template was labeled with Alexa 647 for the visualization of each of the individual components on gel. Cas9 fluorescence signal is shown in green, sgRNA fluorescence signal is shown in red, HDR template fluorescence signal is shown in blue. Reduced mobility indicates charge neutralization and/or complex formation. Position of the components on the gel after the mobility shift assay are indicated with red arrows on the left side of the image. Components were mixed at the indicated molar ratios, and after a short (5 min) incubation time to enable complex formation, samples were loaded onto 1.5% agarose gel and run at 170 V for 15 min. Fluorescent gel images were captured with a ChemiDocTM XRS+ imager (Bio-Rad).

**Figure 3 pharmaceutics-15-02500-f003:**
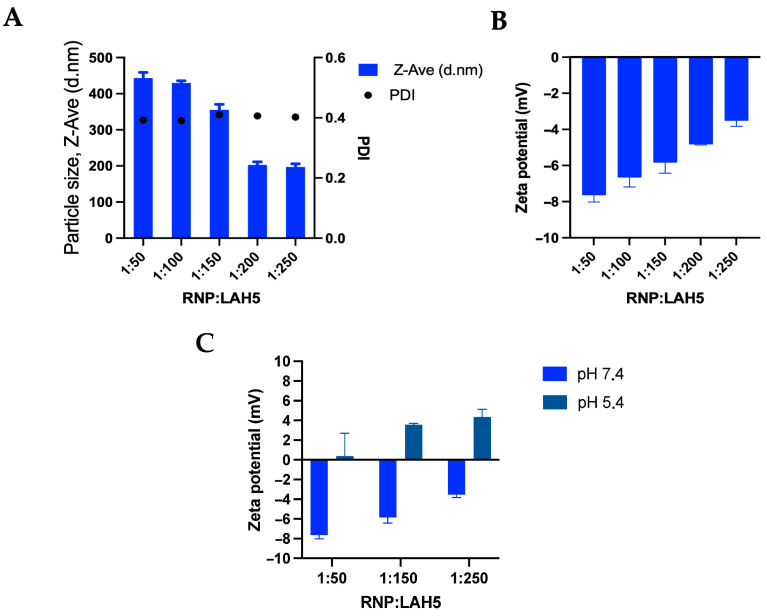
The amphipathic LAH5 peptide forms nanocomplexes with spCas9/sgRNA (RNP). (**A**) Size and polydispersity index of spCas9/sgRNA (RNP) and LAH5 peptide complexation, measuring increasing molar ratios of LAH5 peptide complexed together. (**B**) Zeta potential of spCas9/sgRNA (RNP) and LAH5 peptide complexation, measuring increasing molar ratios of LAH5 peptide complexed together. (**C**) Zeta potential measurements performed at a pH of 7.4 and a pH of 5.4 demonstrate the pH sensitivity of the spCas9/sgRNA (RNP)/LAH5 nanocomplexes. Error bars indicate mean ± SD (*n* = 3).

**Figure 4 pharmaceutics-15-02500-f004:**
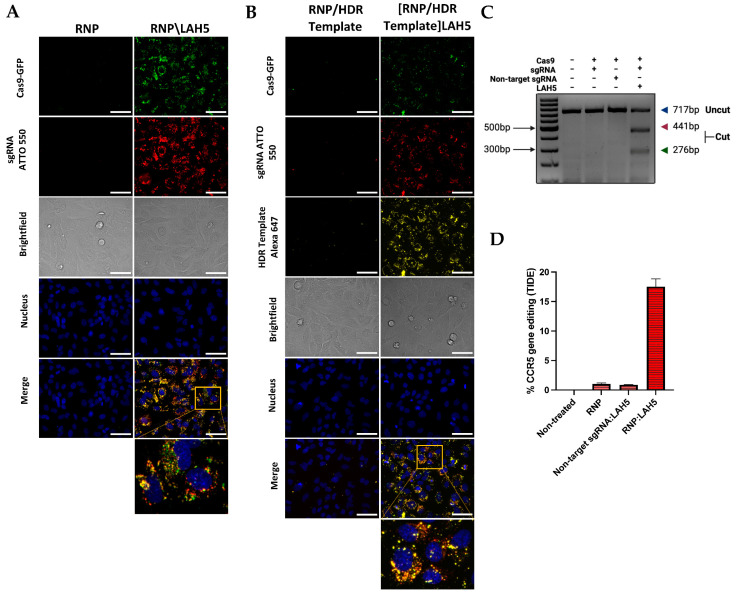
Fluorescence confocal microscopy images of the cellular uptake of Cas9-GFP RNP (**A**) and Cas9-GFP RNP/ssDNA HDR template (**B**) with and without LAH5 complexation. HeLa cells were imaged 24 h after the addition of the indicated components. Scale bars represent 70 μm. Prior to image acquisition, cells were washed 2× with fresh medium to remove excess fluorescently labeled compounds. The concentration of Cas9-GFP RNP was 10 nM for all experiments. The sgRNA that was used was fluorescently labeled with ATTO 550 and ssDNA HDR template with Alexa 647. The merged channels indicate co-localization (yellow) of (**A**) Cas9-GFP and sgRNA ATTO 550 (**B**) Cas9-GFP, sgRNA ATTO 550 and HDR template Alexa 647. Scale bars represent 70 μm. (**C**) T7 endonuclease mismatch detection assay of PCR products containing the CRISPR target. The locus of the *CCR5* gene indicates % *CCR5* gene editing. (**D**) TIDE assay to quantify the frequency of targeted mutations in the target locus of the *CCR5* gene. Data are shown as mean ± SD (*n* = 3).

**Figure 5 pharmaceutics-15-02500-f005:**
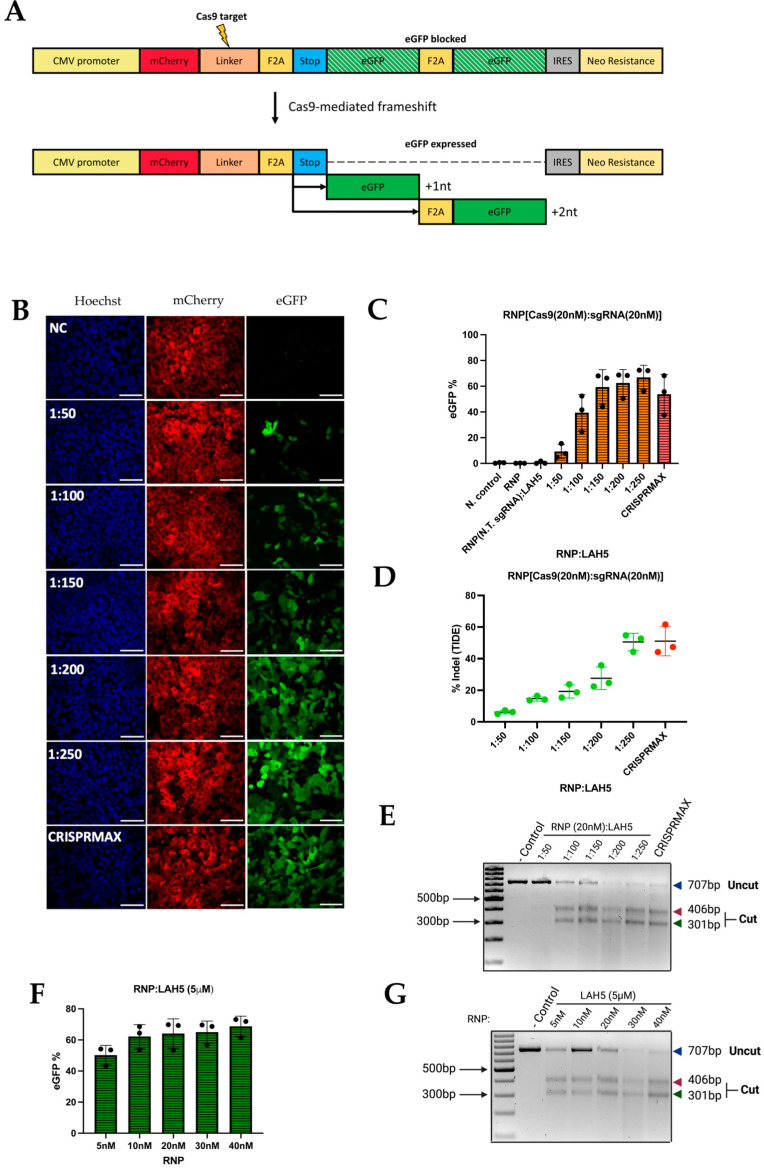
HEK293T stoplight cells were used to determine gene editing efficiency. (**A**) Schematic depiction of the HEK293T stoplight cell reporter construct. These cells stably express the mCherry protein (red) under a CMV promoter (yellow), but eGFP (green) expression is blocked by a stop codon (blue). Cas9-mediated frameshifts in the linker region bypass the stop codon and initiate the production of eGFP upon the introduction of a +1 or +2 frameshift mutation (black arrows). (**B**) Confocal microscopy images (60×) of stoplight HEK293T stoplight cells after treatment with different ratios of RNP:LAH5 peptide at an RNP concentration of 20 nM. Red—mCherry, green—eGFP (Cas9 gene editing), blue—Hoechst 33342 nuclear dye, NC—Negative control. Scale bars represent 70 μm. (**C**) Indel (eGFP positive) cells upon increasing molar ratios of LAH5 peptide (with total 20 nM RNP concentration) measured via flow cytometry. Treatment with LAH5 is indicated in orange, treatment with CRISPRMAX is indicated in red. (**D**) TIDE and (**E**) T7E1 assay performed to detect gene editing of cells transfected with various ratios of RNP/LAH5 peptide nanocomplexes and positive control CRISPRMAX. Treatment with LAH5 is indicated in green, treatment with CRISPRMAX is indicated in red. (**F**) Gene editing efficiency using increasing concentrations of RNP and a stable concentration of LAH5 (5 μM) peptide, gene editing efficiency was analyzed by measuring eGFP(+) cells via flow cytometry. (**G**) T7E1 assay on cells transfected with various ratios of RNP/LAH5 peptide nanocomplexes. Data are presented as mean ± SD (*n* = 3).

**Figure 6 pharmaceutics-15-02500-f006:**
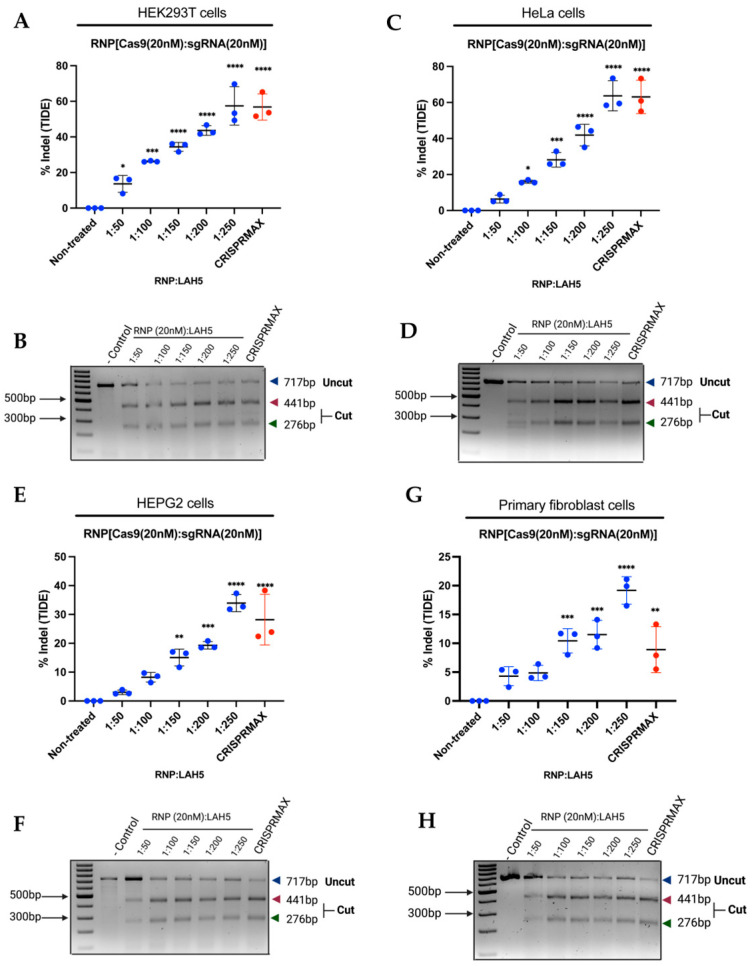
The effectiveness of gene editing was assessed using several cell lines, targeting the *CCR5* gene. (**A**,**B**) Indel measurements in the *CCR5* locus from RNP/LAH5-treated HEK293T cells analyzed via (**A**) TIDE and (**B**) T7E1 assays (uncut, 717 bp; cut, 441 bp 276 bp; Cas9, 20 nM; *CCR5* sgRNA, 20 nM; increasing ratios of LAH5). (**C**,**D**) Indel measurements in the *CCR5* locus from RNP/LAH5-treated HeLa cells analyzed via (**C**) TIDE and (**D**) T7E1 assays. (**E**,**F**) Indel measurements in the *CCR5* locus from RNP/LAH5-treated HEPG2 cells analyzed vI (**E**) TIDE and (**F**) T7E1 assays. (**G**,**H**) Indel measurements in the *CCR5* locus from RNP/LAH5-treated primary fibroblast cells analyzed via (**G**) TIDE and (**H**) T7E1 assays. Differences between the datapoints are represented as mean ± SD (*n* = 3). Treatment with LAH5 is indicated in blue, treatment with CRISPRMAX is indicated in red. Treated samples were compared with non-treated controls using an analysis of variance (ANOVA) with Dunnett’s multiple comparisons test (* *p* < 0.05; ** *p* < 0.01; *** *p* < 0.001; **** *p* < 0.0001).

**Figure 7 pharmaceutics-15-02500-f007:**
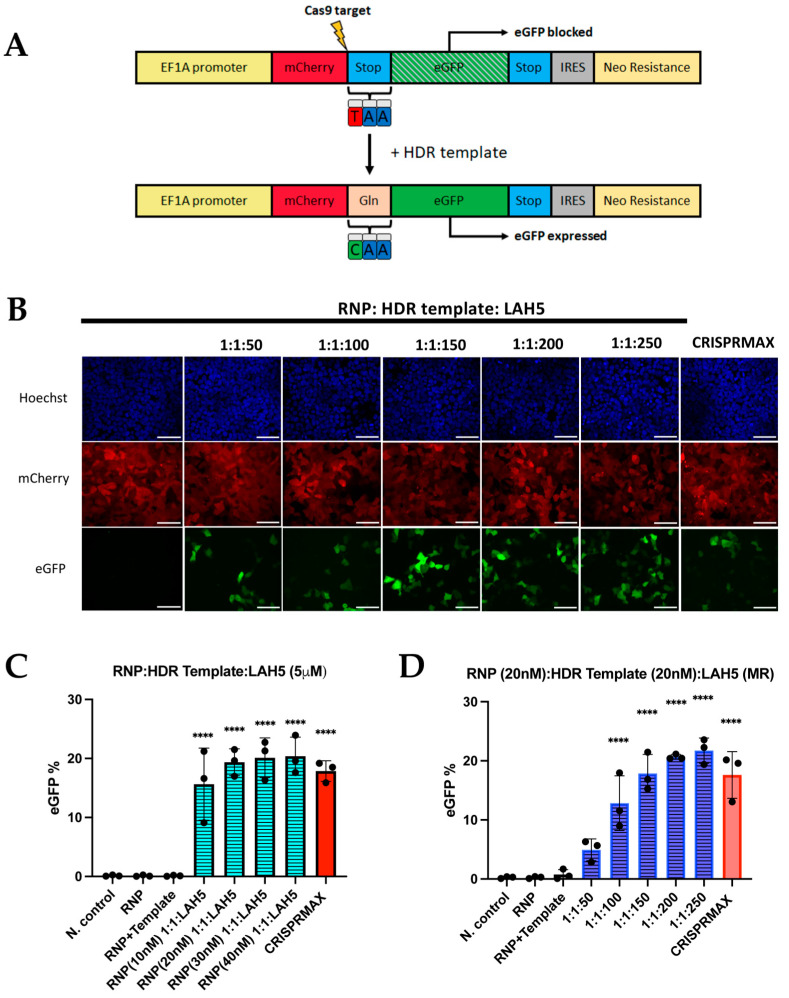
LAH5-peptide-mediated gene correction efficiency tested using the HDR stoplight reporter system. (**A**) Schematic depiction illustrating the mechanism of the construct. HEK293T HDR stoplight cells exhibited continuous expression of mCherry (red), directly followed by a stop codon (blue) and an in-frame open reading frame (ORF) for eGFP (green). As translation stops at the stop codon at the end of the mCherry ORF, eGFP expression is blocked. Introduction of a single-stranded oligonucleotide (ssODN) HDR template along with CRISPR-Cas9 facilitates targeted substitution of the stop codon with a glutamine residue (pink), thereby inducing the synthesis of a fusion protein of mCherry and eGFP. (**B**) Confocal microscopy images (60×) of HDR stoplight HEK293T cells after treatment with RNP (20nM)/HDR template (20 nM) and increasing ratios of LAH5 peptide ranging from 1:1:50 to 1:1:250, and CRISPRMAX as the positive control. Red—mCherry, green—eGFP (gene correction), blue—Hoechst, NC—Negative control. Scale bars represent 70 μm. (**C**) HEK293T HDR stoplight cells were treated with a range of 10 nM to 40 nM RNP/HDR template at a 1:1 ratio complexed with 5 μM LAH5 peptide. HDR efficiency was quantified via flow cytometry by measuring eGFP expression. Treatment with LAH5 is indicated in blue, treatment with CRISPRMAX is indicated in red. (**D**) HEK293T HDR stoplight cells were treated with RNP/HDR template/LAH5 nanocomplexes. The nanocomplexes were prepared by combining RNP (20 nM) and HDR template (20 nM), along with LAH5 peptide, at increasing ratios ranging from 1:1:50 to 1:1:250. HDR efficiency was quantified via flow cytometry by measuring eGFP expression. Treatment with LAH5 is indicated in blue, treatment with CRISPRMAX is indicated in red. Data are presented as mean ± SD (*n* = 3). Treated samples were compared with non-treated controls using an analysis of variance (ANOVA) with Dunnett’s multiple comparisons test (**** *p* < 0.0001).

## Data Availability

The data supporting this research can be found in the Appendix A of this manuscript.

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
