# Peer review of "Amphipathic Cell-Penetrating Peptide-Aided Delivery of Cas9 RNP for In Vitro Gene Editing and Correction"

_pharmaceutics, 2023, doi:10.3390/pharmaceutics15102500_

Round 1

Reviewer 1 Report (New Reviewer)

In this study, researchers have harnessed the potential of the CRISPR/Cas9 gene editing system, overcoming a major hurdle by perfecting a safe and efficient delivery method. Utilizing the LAH5 peptide, the study successfully formed nanocomplexes, allowing seamless intracellular delivery of spCas9:sgRNA ribonucleoprotein (RNP) and RNP/single-stranded homology-directed repair (HDR) template. Notably, the study showcased targeted gene editing of the CCR5 gene across diverse cell lines, underscoring the versatility and promise of the LAH5-based delivery strategy. This research signifies a transformative stride toward the development of tailored CRISPR/Cas-based therapies, offering renewed hope for patients with genetic disorders. However, there are two concerns for this work. 1. In the introduction part, the authors explained why they selected LAH5. I was wondering if they had done studies on any other amphipathic peptides. And will they expect anything different? 2. How is the complexation was characterized besides EMSA in figure 2? Are there any other evidence to support the complexation?

Author Response

Reviewer 2 Report (New Reviewer)

In this manuscript, the authors investigated the applicability of the amphipathic cell-penetrating peptide LAH5 for the intracellular transfer of RNP (spCas9:sgRNA)  and RNP/ssHDR template. They found that the LAH5 peptide (rich in histidine residues) effectively formed nanocomplexes with both RNP and RNP/HDR, and these nanocomplexes showed successful cellular uptake and delivery in vitro, using some cell lines. The topic of their study is interesting and important for the development of delivery systems used in CRISPR/Cas-based genetic engineering. The research results are adequately presented and the article is generally well written, although I found some weaknesses. Here are my specific comments.

11. There is lack of the discussion of the effect labeling of Cas9 by GFP and sgRNA by ATTO-550 on the formation of Cas9-sgRNA and Cas9-sgRNA-LAH5 complexes.  (Results, paragraph 3.1) Similarly, the influence of labeling of HDR by Alexa-647 on the formed complexes.

22. Paragraph 3.2. Only the results are presented here without comments. It would be interesting if the authors had suggestion as to why the size of particle decreases as  the ratio RNP:LAH5 increase (Figure 3A) – this also not discussed in Discussion section. Also, the zeta potential (Figure 3B) increase (or maybe is less “minus) when the ratio is higher and this result should also be  commented.

33. The authors did not include the pH value in the zeta potential experiment shown in Figure 3B, although they discuss it in the Discussion.

44.  In general, there is some chaos in the description of Nanocomplex size and zeta potential results – the results (line 325) are not consistent with the discussion where additional data are presented (lines 610-623).

55. In Figure 5 (HEK293T stoplight cells): A schematic representation of the construct and mechanism used, similar to Figure 7A (HDR Stoplight Reporter System) would be very helpful. The article would also be more consistent in its presentation.

66.  I suggest placing Figure S7 along with Figure 3B as it is an important result and both together clarify the authors' thought.

77. Line 353: Please add “late endosome marker” near CD63-eGFP as done in supplemental Figure S8 describing this experiment

88. The figures are too far  from where they were discussed.

99. Please change the caption of Figure 4 because it is chaotic (e.g., the description of D is before B and C)

110.   A more detailed description is needed when mCherry first appears in the results (line 401): please add “fluorescent protein”

111.   I do not understand what “INDEL” means (lines 404, 457, etc…)

Round 2

Reviewer 2 Report (New Reviewer)

The authors have revised the manuscript in accordance with my comments. I have no further comments.

This manuscript is a resubmission of an earlier submission. The following is a list of the peer review reports and author responses from that submission.

Round 1

Reviewer 1 Report

This paper investigated the delivery ability of Cas9 RNP mediated by LAH5. The expression of fluorescent Cas9 reporter in different ratios of RNP/LAH5 in HEK293T cells was texted. The authors found that LAH5 peptide could efficiently deliver Cas9 RNP with and without HDR template. In the paper, only one positive control CRISPRMAX was used in the transfection experiment. The most common commercial transfection agent, Lipofectamine2000 or PEI, was not compared with LAH5. Besides, LAH5's ability to deliver Cas9 RNP has not been experimentally demonstrated in vivo. Most importantly, the potential of LAH5 as a non-viral gene delivery system has been studied in previous paper (High-content screening of peptide-based non-viral gene delivery systems. Journal of Controlled Release. 2012, 158 ,433–442). The delivery of DNA (Cationic amphipathic histidine-rich peptides for gene delivery. Biochim Biophys Acta, 2006, 1758(3):301-307.) and RNA (Design and evaluation of histidine-rich amphipathic peptides for siRNA delivery. Pharm Res. 2010, 27(7):1426-1436.) by the analogue of LAH5, LAH4, has also been reported, seriously affecting the innovation of this paper. Therefore, I think this paper is not suitable for publication in Pharmaceutics.

Minor editing of English language required

Author Response

This paper investigated the delivery ability of Cas9 RNP mediated by LAH5. The expression of fluorescent Cas9 reporter in different ratios of RNP/LAH5 in HEK293T cells was texted. The authors found that LAH5 peptide could efficiently deliver Cas9 RNP with and without HDR template.

Questions from 1st reviewer;

  1. In the paper, only one positive control CRISPRMAX was used in the transfection experiment. The most common commercial transfection agent, Lipofectamine2000 or PEI, was not compared with LAH5.

Answer: This study primarily focuses on the delivery of Cas9 ribonucleoproteins (RNP), rather than utilizing the plasmid or mRNA format of Cas9. Hence, CRISPRMAX was chosen as the positive control because this lipofectamine-based reagent is specifically optimized for Cas9 RNP delivery and thus a better positive control than the proposed Lipofectamine2000 or PEI. (Yu, X., Liang, X., Xie, H. et al. (2016) Improved delivery of Cas9 protein/gRNA complexes using lipofectamine CRISPRMAX. Biotechnol Lett 38, 919–929. https://doi.org/10.1007/s10529-016-2064-9; Zhang, S., Shen, J., Li, D., Cheng, Y. (2021). Strategies in the delivery of Cas9 ribonucleoprotein for CRISPR/Cas9 genome editing. Theranostics, 11(2), 614-648. https://doi.org/10.7150/thno.47007).  

Yu, X., Liang, X., Xie, H. et al. Improved delivery of Cas9 protein/gRNA complexes using lipofectamine CRISPRMAX. Biotechnol Lett 38, 919–929 (2016). https://doi-org.proxy.library.uu.nl/10.1007/s10529-016-2064-9

  1. Besides, LAH5's ability to deliver Cas9 RNP has not been experimentally demonstrated in vivo.

Answer: We share the curiosity with the reviewer to find out if the LAH5 peptide is suitable for delivery of CRISPR-Cas9 RNP in vivo after systemic or local administration. However, since this cell-penetrating peptide had never been tested for Cas9 RNP delivery, the scope of this research was to investigate if this would indeed be possible and to find optimal conditions for LAH5-mediated delivery on cells in culture. At this stage, we can only speculate if this LAH5 peptide would be suitable for local or systemic in vivo applications, which we have also addressed in the discussion, as highlighted in lines 632-633. Furthermore, we acknowledged the necessity for future experiments to evaluate the stability of the LAH5 peptide-RNP complex under simulated in vivo conditions, as stated in lines 635-637. The discussion section concludes by affirming that this delivery strategy holds potential for ex vivo applications (lines 637-639). Future research will certainly address the in vivo applicability.

  1. Most importantly, the potential of LAH5 as a non-viral gene delivery system has been studied in previous paper (High-content screening of peptide-based non-viral gene delivery systems. Journal of Controlled Release. 2012, 158 ,433–442). The delivery of DNA (Cationic amphipathic histidine-rich peptides for gene delivery. Biochim Biophys Acta, 2006, 1758(3):301-307.) and RNA (Design and evaluation of histidine-rich amphipathic peptides for siRNA delivery. Pharm Res. 2010, 27(7):1426-1436.) by the analogue of LAH5, LAH4, has also been reported, seriously affecting the innovation of this paper. Therefore, I think this paper is not suitable for publication in Pharmaceutics.

Answer: We regret our failure in effectively conveying the novelty of our manuscript. We acknowledge the importance of the suggested studies, which served as inspiration for our research. However, it is crucial to note that the application of LAH5 or similar peptides in prior reports involved the delivery of distinct cargoes, such as plasmids in the first report and siRNA in the second. In contrast, our study focuses on the utilization of LAH5 for the targeted delivery of sgRNA:Cas9 ribonucleoproteins (RNP) to facilitate gene editing and gene correction. Remarkably, the use of LAH5 peptide for Cas9 RNP delivery or therapeutic protein delivery applications has not been explored previously. Moreover, the significance of our study lies in demonstrating that LAH5 can form a nanocomplex incorporating multiple components, including Cas9 protein, sgRNA, and a ssDNA HDR template (illustrated in Figure 2 through the mobility shift assay). The efficient delivery of these components in vitro allows us to observe unprecedented functional effects, as evidenced by the results presented in Figures 5, 6, and 7.

To further emphasize the novelty of using LAH5 for Cas9 RNP delivery, we have changed a few sentences in the introduction:

“In this study, the aim was to determine if LAH5 would also be suitable for delivery of Cas9 RNP to a variety of cells in culture. The use of LAH5 peptide for Cas9 RNP delivery or therapeutic protein delivery applications has not been explored previously.  We show for the first time that LAH5 peptides form nanocomplexes with Cas9 RNP that were efficiently internalized by a variety of different cell lines.”

Reviewer 2 Report

1. In Summary this sentence need to be improved 

 Peptide/RNP 16 nanocomplexes were characterized for size and charge, and nanocomplexation with RNP and 17 RNP/HDR cargo was demonstrated via an electrophoretic mobility shift assay. 

2. Summary must be improved

3. Sentence must be improved: 

 Viral vectors can 34 be constructed that encode the Cas9 protein along with a single guide RNA for 35 introducing targeted double strand breaks in the genome, but the enduring expression of 36 Cas9 in the target cell population may lead to undesired off-target effects as well as 37 potential immunogenicity 

4. What do you mean with this sentence: that might be toxic to primary cells 75 and has limited physiological relevance. 

5. Which is the context to modify the CCR5 gene ?

6. Space between lines are different in paragrphs.

7. Letter format is also different

8. You have to explain why is important to measure  size distribution and the zeta potential 

9. Which is the impact in cells that nanoparticles are positive or negative?

10. Why the funding is coming form Turkey?

English should be revised.

Author Response

  1. In Summary this sentence needs to be improved 

 Peptide/RNP 16 nanocomplexes were characterized for size and charge, and nanocomplexation with RNP and 17 RNP/HDR cargo was demonstrated via an electrophoretic mobility shift assay. 

Answer: We are thankful for this valuable comment. The sentence was rewritten as follows: ” Furthermore, by using an electrophoretic mobility shift assay with fluorescently-labelled HDR template, sgRNA and Cas9 protein we were able to demonstrate that all components were present in the nanocomplexes.”

We have highlighted these sentences in the revised manuscript with yellow.

  1. Summary must be improved

Answer: Summary has been rewritten.

The therapeutic potential of the CRISPR/Cas9 gene editing system in treating numerous genetic disorders is immense. To fully realize this potential, it is crucial to achieve safe and efficient delivery of CRISPR-Cas9 components into the nuclei of target cells. In this study, we investigated the applicability of the amphipathic cell-penetrating peptide LAH5, previously employed for DNA delivery, in intracellular delivery of spCas9:sgRNA ribonucleoprotein (RNP) and RNP/single-stranded homology-directed repair (HDR) template. Our findings revealed that the LAH5 peptide effectively formed nanocomplexes with both RNP and RNP/HDR cargo, and these nanocomplexes demonstrated successful cellular uptake, and cargo delivery. Furthermore, by using an electrophoretic mobility shift assay with fluorescently-labelled HDR template, sgRNA and Cas9 protein we were able to demonstrate that all components were present in the nanocomplexes.  Functional screening of various ratios of peptide/RNP nanocomplexes was performed on fluorescent reporter cell lines to assess gene editing and HDR-mediated gene correction. Moreover, targeted gene editing of the CCR5 gene was successfully demonstrated across diverse cell lines. This LAH5-based delivery strategy represents a significant advancement toward the development of therapeutic delivery systems for CRISPR/Cas-based genetic engineering.

  1. Sentence must be improved: 

Viral vectors can 34 be constructed that encode the Cas9 protein along with a single guide RNA for 35 introducing targeted double strand breaks in the genome, but the enduring expression of 36 Cas9 in the target cell population may lead to undesired off-target effects as well as 37 potential immunogenicity. 

Answer: Thank you for the comment, the sentence was rewritten, and the new sentence was highlighted in the main text.

“viral vectors can be constructed to carry the genetic information to encode for the Cas9 protein as well as the single guide RNA as a means of intracellular delivery of CRISPR-Cas. However, since expression from viral promoters is often strong and long-lasting, this can give rise to unintended off-target effects and potential immunogenic responses.”

  1. What do you mean with this sentence: that might be toxic to primary cells 75 and has limited physiological relevance. The system did not tested on broad range of primary cells therefore it can cause some toxicity and we cannot observe same efficiency as we detected on reporter cells.

Answer: we agree with the reviewer that the sentence was not clear. We have therefore rewritten it as follows:

“Important to note is that these efficiencies were reached under conditions favoring optimal transfection efficiencies (use of easy to transfect HeLa and HEK293T reporter cell lines, medium without serum and addition of 5% PEG-PVA), that will be difficult to replicate under physiologically relevant conditions.”

We have highlighted the change in yellow in the main text.

  1. Which is the context to modify the CCR5 gene?

Answer: Besides showing LAH5-mediated gene editing in a reporter cell line, we wanted to show gene editing on an endogenous gene locus. We have selected the CCR5 gene as an endogenous target because it exerts minimal impact on the phenotype of the targeted cells.

  1. Space between lines are different in paragraphs.

Thank you for the comment. The space between lines checked and spaces were adjusted.

  1. Letter format is also different

Thank you for your comment. We have changed the font and font size accordingly.

  1. You have to explain why is important to measure size distribution and the zeta potential 

Answer: This is important to assess colloidal stability of aqueous dispersions of nanoparticles, including supramolecular nanocomplexes, which is defined by the Derjaguin/Landau/Verwey/Overbeek (DLVO) theory. High negative or positive zeta-potentials and monodisperse particles whose size do not increase over time are indications of stable colloid dispersions. A colloidal stable nanocomplex is needed under storage conditions to prevent aggregation or agglomeration, but also for cellular uptake, which is highly dependent on the nanoparticle size.   

In the discussion section we explained the impact of size and zeta potential by these sentences “ Even though hydrophobic clustering of the LAH5 peptide itself cannot be excluded, as was demonstrated before [37], and the formed nanocomplexes are polydisperse, we observed stable size of the nanocomplexes at 1:200 and 1:250 RNP/LAH5 ratios and confirmed this interaction via EMSA assay. This offers the opportunity to efficiently deliver multiple components needed for gene editing with a single delivery system. To show proof-of-concept for this, we have demonstrated that the co-delivery of a ssDNA template with the Cas9 RNP led to targeted gene correction in a newly developed reporter cell line.”

  1. Which is the impact in cells that nanoparticles are positive or negative?

Answer:

Since the Cas9 RNP has a net negative charge, an excess of positive charges is desired for making stable nanocomplexes. However, CPPs containing many positive charges, such as R11 are often quite toxic to cells as the positive charges interfere with the integrity of cell membranes. Therefore, it is important to precisely balance the overall amount of positive charges within the nanocomplexes to facilitate complexation, but to limit cytotoxicity as much as possible. The LAH5 peptide is suitable for this as the overall positive charges can be adjusted by change in pH, by virtue of the pkA of the histidine groups present in the peptide.  Cas9 RNP: LAH5 nanocomplexes exhibit a negative charge at pH 7.4. However, when the pH level decreases to 5.4, the nanocomplexes acquire a positive charge (data shown below). This attribute enables the RNP: LAH5 nanocomplexes to effectively disrupt endosomal membranes due to the pH imbalance. Notably, the pH level undergoes changes during endosomal uptake, and the acidic environment within the endosome facilitates this phenomenon.

In discussion section  line 612-617 these sentences were added “Moreover, at a pH level of 5.4, the nanocomplexes undergo a shift towards a positive charge (Figure S7). This characteristic empowers the RNP: LAH5 nanocomplexes to proficiently interfere with endosomal membranes, capitalizing on the pH disparity. It is worth emphasizing that the pH level experiences fluctuations during the process of endosomal uptake, and the acidic milieu within the endosome plays a pivotal role in facilitating this phenomenon.”

To support this finding we added additional zeta potential measurement data in the supplementary file as figure s7.

Zeta potential data

  1. Why the funding is coming from Turkey?

Answer: Mert Öktem is a recipient of a scholarship awarded by the Turkish Ministry of Education.

Reviewer 3 Report

Cell-penetrating peptides (CPPs) have been extensively explored for their capacity to deliver Cas9 RNPs in various cell types mainly in HeLa or HEK293T reporter cell lines, known for ease of transfection. Herein LAH5 (KKALLALALHHLAHLAH HLALALKKA) is shown to have the capacity to deliver at the same time several components (RNP and HDR templates), which are crucial for CRISPR-mediated gene correction.

Specific comments

- A major concern is about the material being used for the transfection. There is a  significant lack of information regarding its composition and topology.

1-Composition.We are dealing with a three-components system and one can rule out the possibility that the three components will assemble with a 100% yield. This means that the real proportion of each element in the complex used in the transfection experiments is unknown. This lack of information raises serious questions regarding the interpretation of the data and the quality control of the sample which will be a prerequisite for any clinical trial.

2 -Topology. No information is available regarding the orientation of the components in the LAH5-RNP-HDR complex. Which species is pointing at the surface of the complexes. What is the composition of the core. A central core made of a hydrophobic clustering of LAH5 surrounded with the Cas9 RNP material is a possibility that should be confirmed. An exact knowledge of the proportions of LAH5, Cas9RNP and HDR should allow to build a model compatible with a 200nm-400nm size.

-The punctuate nature of the intracellular fluorescence points towards endocytic uptake. However there are no obvious reasons to rule out a direct interaction of the LAH5-RNP with the plasma membrane. An outwards orientation of the LAH5 peptides would facilitate a destabilization of the plasma membrane.

- What is the stability of LAH5:Cas9 RNP nanocomplexes at endosomal PH. Dissociation of the nanocomplexes would favor the LAH5 induced-destabilisation of the endosomal membrane via a carpet-like model

- I don’t share the pessimistic view of the authors who think that the current nano-complex formulation is not stable enough for direct in vivo applications. Is it some experimental evidence suggesting that intravenous injection quickly dilutes out the free peptide.

Author Response

Cell-penetrating peptides (CPPs) have been extensively explored for their capacity to deliver Cas9 RNPs in various cell types mainly in HeLa or HEK293T reporter cell lines, known for ease of transfection. Herein LAH5 (KKALLALALHHLAHLAH HLALALKKA) is shown to have the capacity to deliver at the same time several components (RNP and HDR templates), which are crucial for CRISPR-mediated gene correction.

Specific comments

- A major concern is about the material being used for the transfection. There is a significant lack of information regarding its composition and topology.

1-Composition.We are dealing with a three-components system and one can rule out the possibility that the three components will assemble with a 100% yield. This means that the real proportion of each element in the complex used in the transfection experiments is unknown. This lack of information raises serious questions regarding the interpretation of the data and the quality control of the sample which will be a prerequisite for any clinical trial.

Answer: This is a very valuable comment and we fully agree with the reviewer that a more in-depth physicochemical characterization of the nanocomplexes is needed before this system can be brought into the clinic. At this stage, we have no quantitative data of the proportion of each component in the nanocomplexes. We only know that each component is present inside the nanocomplexes as determined by gel retardation of fluorescently-labelled components and that this leads to functional delivery. To get trustworthy quantitative data, analytical methods should be developed that do not require separation of the nanocomplexes from the free components (as this would result in a shift in equilibrium and hence biased data), and that can detect all three components in their native state without relying on fluorescence. This would be a study on its own and falls outside the scope of this research in which we aimed to demonstrate proof-of-concept for the use of LAH5 peptide for Cas9 RNP delivery.

We have added a few sentences to the discussion part to emphasize the urge for more in-depth physicochemical characterization of the LAH5-Cas9 RNP nanocomplexes.

“It is important to note that further biophysical and structural characterization of the LAH5 nanocomplexes is needed to elucidate the stoichiometry of each of the components in the nanocomplexes as well as the topology of the supramolecular multicomponent assemblies. It is at present unclear what the dynamics of release are of each of the cargos and if LAH5 is accessible on the surface of the nanocomplexes for membrane interaction or that it needs to be release first in order to be membrane active. Nevertheless, having demonstrated that multiple components can be complexed by LAH5, it offers the opportunity to efficiently deliver multiple components needed for gene editing with a single delivery system.”

2 -Topology. No information is available regarding the orientation of the components in the LAH5-RNP-HDR complex. Which species is pointing at the surface of the complexes. What is the composition of the core. A central core made of a hydrophobic clustering of LAH5 surrounded with the Cas9 RNP material is a possibility that should be confirmed. An exact knowledge of the proportions of LAH5, Cas9RNP and HDR should allow to build a model compatible with a 200nm-400nm size.

Answer: Thank you very much for this comment. Same as the previous comment, the suggested in-depth characterization of the nanocomplexes is needed. The discussion was adapted to reflect this.

3 -The punctuate nature of the intracellular fluorescence points towards endocytic uptake. However, there are no obvious reasons to rule out a direct interaction of the LAH5-RNP with the plasma membrane. An outwards orientation of the LAH5 peptides would facilitate a destabilization of the plasma membrane.

Answer: We agree that direct translocation cannot be ruled out, which has been described for arginine-rich CPPs such as R9 and R11 (Schmidt, N.; Mishra, A.; Lai, G. H.; Wong, G. C. L. FEBS Lett. 2010, 584, 1806–1813. doi:10.1016/j.febslet.2009.11.046; Ter-Avetisyan, G.; Tünnemann, G.; Nowak, D.; Nitschke, M.; Herrmann, A.; Drab, M.; Cardoso, M. C. J. Biol. Chem. 2009, 284, 3370–3378. doi:10.1074/jbc.m805550200). However, this mode of cell penetration is mostly observed at rather high concentrations of CPPs, is highly cell-type dependent and is severely limited when large cargos are attached to the CPPs. Moreover, several studies show that cytoplasmic localization of CPPs was abolished when inhibitors of endocytosis were applied (Fischer R, Köhler K, Fotin-Mleczek M, Brock R (2004). A stepwise dissection of the intracellular fate of cationic cell-penetrating peptides. J Biol Chem 279:12625–12635; Liu BR, Li J-F, Lu S-W, et al (2010) Cellular internalization of quantum dots noncovalently conjugated with arginine-rich cell-penetrating peptides. J Nanosci Nanotechnol 10:6534–6543). Hence, we believe that the major route, leading to functional delivery of Cas9 RNP is via endocytosis, with subsequent pH-dependent endosomal membrane destabilization.  

We have changed the sentence (Line 345-347) in the results section on cellular uptake to clarify this point.

“The punctuate nature of the intracellular fluorescence points towards endocytic uptake, even though low levels of direct membrane translocation cannot be excluded.”

4 - What is the stability of LAH5:Cas9 RNP nanocomplexes at endosomal PH. Dissociation of the nanocomplexes would favour the LAH5 induced-destabilisation of the endosomal membrane via a carpet-like model?

Answer: Thank you very much for this comment. The stability of RNP/LAH5 nanocomplexes at endosomal pH was not assessed in this study. However, the zeta potential measurements at different pH levels (as depicted in the graph below) clearly demonstrate the pH-dependent effect of the amphipathic LAH5 peptide. Additionally, the in vitro data on gene editing and gene correction provide compelling evidence for the superior endosomal escape capability of RNP/LAH5 nanocomplexes. Nevertheless, there is currently no specific information available about the exact mechanism by which these peptides destabilize endosomal membranes. The amphipathic nature of the LAH5 peptide could support the carpet-like model, but as already pointed out, we have no direct evidence for this. Further biophysical research is needed to elucidate this.

5 - I don’t share the pessimistic view of the authors who think that the current nano-complex formulation is not stable enough for direct in vivo applications. Is it some experimental evidence suggesting that intravenous injection quickly dilutes out the free peptide.

Answer: All transfection experiments conducted in this study were carried out in Opti-MEM with reduced serum content. Although not presented here, additional experiments were performed, demonstrating a clear dependence of gene editing and correction efficiency on serum content. We have no data on the stability of the formed nanocomplexes upon dilution, but the relatively large excess of LAH5 peptide that is needed to form the nanocomplexes hints towards weak interactions. For these reasons, we doubt these nanocomplexes will be suitable for intravenous application. We are therefore certainly planning such in vivo experiments for the near feature to either confirm or reject this hypothesis.

Reviewer 4 Report

In this paper Öktem et al report the use of CPP LAH5 as cell delivery vehicle via nanoparticle formation into mammalian cells for a CRISPR Cas9/sgRNA complex, and demonstrate effective gene editing using this strategy. Overall this is a scientifically sound and interesting paper.

However, some issues should be addressed prior to publication:

Specific points:

1.       Lines 60-62: For most CPPs, the endosomal release is very slow and is the rate limiting process.

2.       Line 323: In general intracellular delivery (of nucleic acids) by CPPs, as well as cationic lipids are dependent on a positive zeta potential responsible for binding to the negatively charged cell surface (leading to endosomal uptake), as also mentioned on page 2. Nonetheless, the RNP-LAH5 particles used here have a negative Zeta potential, and thus seems to be taken up by another mechanism! This requires some explanation.

3.       The fluorescence confocal microscopy images presented to document cellular uptake clearly show uptake and also co-localization of Cas9 and sgRNA (as well as HDR template (Fig. 4 A & D). However, the staining is very inhomogeneous, and apart from an indication that very little is in the nucleus, the images are of so poor resolution that it is not at all possible to assign any location (e.g. endomal, nuclear etc) to the delivered material. Much better data must be provided, not least since these could provide information about the mechanism of uptake.

4.       It seems surprising that practically no dose response is seen for gene editing activity (Fig. 5E)

5.       Cytotoxicity is only studied at the low concentrations used in the in vitro cell assays (Fig. S1). The present delivery method is only of broad interest if it eventually can be used for in vivo application. Therefore, the cell toxicity should be studied at much higher (>10 fold) concentration in order to be relevant for the potential of the delivery method.

Please edit carefully for typographical errors etc.

Author Response

In this paper Öktem et al report the use of CPP LAH5 as cell delivery vehicle via nanoparticle formation into mammalian cells for a CRISPR Cas9/sgRNA complex and demonstrate effective gene editing using this strategy. Overall, this is a scientifically sound and interesting paper.

However, some issues should be addressed prior to publication:

Specific points:

  1. Lines 60-62: For most CPPs, the endosomal release is very slow and is the rate limiting process.

Answer: We appreciate the additional comment suggested by the reviewer. However, given the reported variety of both mechanisms and levels of efficiency of different CPPs, we find this statement a bit too strong to directly include into the manuscript. However, we agree with the reviewer that the importance of endosomal escape as a general rate-limiting factor for macromolecule delivery should be addressed. As a compromise, we included the following sentence on line 64-66:   “ Within the endocytic compartments the CPPs interact with the endosomal membranes, often triggered by change of pH in the endosomes, which results in endosome destabilization and partial cargo release [6, 26].”

  1. Line 323: In general, intracellular delivery (of nucleic acids) by CPPs, as well as cationic lipids are dependent on a positive zeta potential responsible for binding to the negatively charged cell surface (leading to endosomal uptake), as also mentioned on page 2. Nonetheless, the RNP-LAH5 particles used here have a negative Zeta potential, and thus seems to be taken up by another mechanism! This requires some explanation.

Answer: The reviewer raises an important point regarding the functionality of CPPs, and the effect of not only their charge – but also the charge of a nanoparticle on cellular uptake. Indeed, a positive charge of nanoparticles leads to efficient endosomal uptake. However other factors such as charge density and hydrophobicity are of equal importance (E Frohlich, International Journal of Nanomedicine 2012)..

To address this comment, we added the following sentences to the discussion section:

Interestingly, despite a negative zeta potential we observed fast uptake and delivery of our LAH5-RNP complexes, demonstrating efficient cargo delivery after cell exposure of only 3 hours (Figure S3). Whereas a negative charge of nanoparticles generally leads to decreased endosomal uptake, other factors such as charge density and hydrophobicity are of equal importance and may explain these data [E Frohlich, International Journal of Nanomedicine 2012]. Work by A Kichler et al showed through the use of (ATR)-FTIR spectroscopy that the LAH4 family exhibit alpha-helical confirmations with all positive histidine residues located on one side of the helix [A Kichler, PNAS, 2003]. Further NMR Spectroscopy indicated that interaction of these CPPs with membranes were modulated by these histidine-rich side-chains.

  1. The fluorescence confocal microscopy images presented to document cellular uptake clearly show uptake and also co-localization of Cas9 and sgRNA (as well as HDR template (Fig. 4 A & D). However, the staining is very inhomogeneous, and apart from an indication that very little is in the nucleus, the images are of so poor resolution that it is not at all possible to assign any location (e.g. endomal, nuclear etc) to the delivered material. Much better data must be provided, not least since these could provide information about the mechanism of uptake.

Answer: We agree with the reviewer that the included images were of insufficient resolution to allow the interpretation of cellular localization of the various cargos. This issue was the result of unintended image compression upon exporting the final figure. We have addressed this issue, and included high resolution TIFF files of each figure. 

Regarding the inhomogeneous signal distribution, this is the result of cellular uptake and trafficking of the delivered cargo: The detected fluorescent signals in the images are emitted by the labeled components. Consequently, certain components remain trapped within the endosome, observable as punctuated structures, while others manage to escape. By adjusting the fluorescence settings, we can enhance the intensity of the fluorescent signal. However, in this particular case, only the highly intense uptake is visible in the images. Therefore, the images presented have been optimized using these specific settings.

  1. It seems surprising that practically no dose response is seen for gene editing activity (Fig. 5E)

Answer: Indeed, whereas an increase in gene editing is seen from 5 nM to 20 nM, a plateau is reached at higher doses. The inability to detect a dose-response can likely be attributed to the saturation of the reporter system, as the fluorescent reporter system will only show eGFP expression in case of +1nt or +2nt frameshift [De Jong, Nat Commun 2020]. If lower concentrations of RNP had been tested, it is plausible that significant differences between the doses would have been observable. Regrettably, this study did not include testing RNP concentrations lower than 5 nM.

For clarity, the following statement has been added to the manuscript: At these concentrations, cell viability was >80% as compared to control conditions, and the maximum theoretical activation of the fluorescent stoplight reporter was approached [De Jong, Nat Commun 2020].

  1. Cytotoxicity is only studied at the low concentrations used in the in vitro cell assays (Fig. S1). The present delivery method is only of broad interest if it eventually can be used for in vivo application. Therefore, the cell toxicity should be studied at much higher (>10 fold) concentration in order to be relevant for the potential of the delivery method.

Answer: We thank the reviewer for this comment. The results obtained from the cytotoxicity assay revealed that when the RNP:LAH5 peptide ratio exceeded 1:400, toxicity was observed across multiple cell types. Furthermore, optimal concentrations of RNP:LAH5 peptide were determined and validated in various cell types. Cytotoxicity experiments were conducted in 96-well plates, and the nanocomplexes were tested at different concentrations on approximately 10,000 cells, varying depending on the specific cell type. It is evident that testing significantly higher doses would notably decrease cell viability. Generally, in vivo, experiments involve dose calculations based on the animal's weight, which often necessitates much higher doses compared to in vitro settings. Thus, we believe that in vitro cytotoxicity experiments can only provide an indication and cannot be directly compared to in vivo conditions. Moreover, as we address the potential limitations of in vivo applications of these particles (Line 631-633), and underline that use of this CPP in its current form is  specifically suitable for ex vivo applications (Line 635-637), we feel that such additional assays to study in vivo applications fall outside of the scope of this study.

Reviewer 5 Report

The present article evaluated the efficacy of an amphipathic cell-penetrating peptide (CPP) in delivering spCAS9:sgRNA RNP and RNP/HDR template intracellularly for gene editing and correction. This is important since sustained expression of CAS9 in the case of viral vector-based CRISPR/CAS9 strategy may cause unintended side effects whereas a transient Cas9 protein delivery will do the gene editing and then will be degraded. Moreover, the gene editing process will be much faster as no transcription or translation of Cas9 protein will be necessary. Here the authors used a previously published CPP (LAH5) for overcoming the challenges associated with the intracellular delivery of spCAS9:sgRNA RNP and RNP/HDR template.

The idea of using CPP for delivering therapeutic cargo in general and Cas9 RNP in particular is not novel as several studies have evaluated the same in recent years. However, this study is of interest since it demonstrated the use of a CPP in delivering three different components (Cas9 protein, sgRNA, and ssDNA HDR template) successfully for gene editing and correction. Overall, this is a well-executed study with the results supporting the conclusions. However, there are some concerns that should be addressed as outlined below. 

1.       As suggested by the authors in the discussion section, this is suitable for in vitro or ex vivo gene editing/correction only. The authors did not check serum stability or in vivo application of these nano complexes, which dilutes the significance of this study to some extent.

2.       HDR efficiency is still suboptimal at ~20% in some cells specially in primary cells. How does this compare with viral vector-based delivery?

3.       Please include appropriate reference at page 21 line 551.

Author Response

The present article evaluated the efficacy of an amphipathic cell-penetrating peptide (CPP) in delivering spCAS9:sgRNA RNP and RNP/HDR template intracellularly for gene editing and correction. This is important since sustained expression of CAS9 in the case of viral vector based CRISPR/CAS9 strategy may cause unintended side effects whereas a transient Cas9 protein delivery will do the gene editing and then will be degraded. Moreover, the gene editing process will be much faster as no transcription or translation of Cas9 protein will be necessary. Here the authors used a previously published CPP (LAH5) for overcoming the challenges associated with the intracellular delivery of spCAS9:sgRNA RNP and RNP/HDR template.

The idea of using CPP for delivering therapeutic cargo in general and Cas9 RNP in particular is not novel as several studies have evaluated the same in recent years. However, this study is of interest since it demonstrated the use of a CPP in delivering three different components (Cas9 protein, sgRNA, and ssDNA HDR template) successfully for gene editing and correction. Overall, this is a well-executed study with the results supporting the conclusions. However, there are some concerns that should be addressed as outlined below. 

  1. As suggested by the authors in the discussion section, this is suitable for in vitro or ex vivo gene editing/correction only. The authors did not check serum stability or in vivo application of these nano complexes, which dilutes the significance of this study to some extent.

Answer: Thank you very much for this comment. While the suggestion to explore LAH5-mediated RNP delivery in vivo is certainly valuable, it is essential to clarify that this study is confined solely to in vitro experiments and does not encompass in vivo investigations. Nonetheless, it is worth mentioning that LAH5 can be conjugated with various modifications, which are currently being investigated in a separate follow-up study. Preliminary data shows that such approaches enable the formation of more stable nanoparticles with RNP, thus opening up potential avenues for utilizing this peptide-mediated delivery strategy in vivo applications.  However, as mentioned for Reviewer 1 and 4 as well, we address the potential limitations of in vivo applications of these particles (Line 631-633), and underline that use of this CPP in its current form is  specifically suitable for ex vivo applications (Line 635-637). Furthermore, we acknowledged the necessity for future experiments to evaluate the stability of the LAH5 peptide-RNP complex under simulated in vivo conditions, as stated in lines 633-635. Future research will certainly address the in vivo applicability.

  1. HDR efficiency is still suboptimal at ~20% in some cells specially in primary cells. How does this compare with viral vector-based delivery?

Answer: Thank you very much for this comment. The process of homology-directed repair (HDR) relies on the cell cycle, specifically during the G2/M phases when the nuclear pore complex is open, facilitating the access of RNP/HDR complexes to the nucleus. Conversely, during other phases of the cell cycle, the nucleus is less receptive to the uptake of DNA templates, resulting in a significant decrease in HDR efficiency. Primary cells, which have a slower division rate compared to HEK 293T and HeLa cells, experience a notable reduction in HDR efficiency. Moreover, whereas there are several studies that have reported higher HDR efficiencies when employing additional strategies such as chemically modified HDR templates, retaining cells in the G2/M phase through chemical or genetic strategies, or using modified endonucleases, 20% HDR efficiency is not uncommon. In fact, one of the most commonly used highly sensitive eGFP to BFP HDR reporter assays by Glasser et al shows a HDR efficiency of 23.3% [Glasser, Mol Ther Nucleic Acids 2016].  

In terms of viral-based delivery, these delivery systems are not reliant on nuclear uptake but rather employ active nuclear trafficking strategies, making direct comparisons of HDR efficiency between synthetic and viral delivery systems challenging. Moreover, it should be noted that direct comparison of viral delivery strategies with a synthetic RNP delivery + HDR template would be additionally difficult to directly compare, as no current viral delivery vector is capable of delivering protein, RNA and a DNA template at once.

Furthermore, as mentioned in the introduction of the comments, it is noteworthy that this study represents the first demonstration of a peptide's ability to deliver RNP/HDR templates together, adding significant value to its findings.

  1. Please include appropriate reference at page 21 line 547.

Answer: Thank you very much for the comment. The reference added and highlighted in the main text.

In this report, we have repurposed the LAH5 amphipathic peptide, initially developed for transfection of pDNA [32], for CRISPR-Cas9 RNP delivery to a variety of human cells in vitro.

Round 2

Reviewer 3 Report

;

Reviewer 4 Report

Although the microscopy images have been graphically improved, the data are not convincing.

Specifically:

1.     In order to assign the fluorescence from the delivered cargos to cellular localizations, which is critical for understanding the uptake pathway and mechanism and thus to discussing the significance of the new cellular delivery protocol, higher confocal microscopic magnification and resolution is required.

2.     In addition, parallel phase contrast images showing the cells and there morphology, which is standard in microscopy, are missing, and required, in order to interpret the fluorescence images. In particular, since the fluorescence intensity and pattern appear to vary considerably between different cells (especially difficult because cell images are missing). Finally, the nuclear staining images, in particular in Fig. 4D  but also 4B, do not seem to align with the fluorescence images!

3.     The authors now conclude from these images (page 10) that “The punctuate nature of the intracellular fluorescence points towards endocytic uptake, even though low levels of direct membrane translocation cannot be excluded”. The present images do not allow this conclusion!

In conclusion, new and better fluorescence confocal microscopy data are required in order for this paper to be acceptable for publication

Other points

4.     On page 2 the authors now write that ”Important to note is that these efficiencies were reached under conditions favoring optimal transfection efficiencies (use of easy to transfect HeLa and HEK293T reporter cell lines, medium without serum and addition of 5% PEG-PVA), that will be difficult to replicate under physiologically relevant conditions”. This fact seriously reduces the general interest and importance of the study.

5.     On page 1 the authors now argue that: ” Viral vectors can 37 be constructed to carry the genetic information to encode for the Cas9 protein as well as the single guide RNA as a means of intracellular delivery of CRISPR-Cas. However, since expression from viral promoters is often strong and long-lasting, this can give rise to unintended off-target effects and potential immunogenic responses [6, 7, 8]. More transient systems are therefore desired”. However, their new system cannot be used under physiological conditions and in vivo. Therefore this argument is not valid.

6.     Likewise in the abstract it is stated that:”This LAH5-based delivery strategy represents a significant advancement toward the development of therapeutic delivery systems for CRISPR/Cas-based genetic engineering”

This is not correct since overall this new delivery system is solely for in vitro applications under non-physiological conditions and  consequently so far has very limited general utility and importance.

Therefore, it must be explicitly stated in the title and the abstract that it is an “in vitro delivery” method.

overall ok